# Earthquake crisis unveils the growth of an incipient continental fault system

Eulàlia Gràcia [1], Ingo Grevemeyer [2], Rafael Bartolomé [1], Hector Perea [1,3], Sara Martínez-Loriente [4], Laura Gómez de la Peña [1,2], Antonio Villaseñor [5], Yann Klinger [6], Claudio Lo Iacono[7], Susana Diez [8], Alcinoe Calahorrano [1], Miquel Camafort[1], Sergio Costa[1], Elia d'Acremont [9], Alain Rabaute [9] & César R. Ranero [1,10]

Large continental faults extend for thousands of kilometres to form boundaries between rigid tectonic blocks. These faults are associated with prominent topographic features and can produce large earthquakes. Here we show the first evidence of a major tectonic structure in its initial-stage, the Al-Idrissi Fault System (AIFS), in the Alboran Sea. Combining bathymetric and seismic reflection data, together with seismological analyses of the 2016 $M_w$ 6.4 earthquake offshore Morocco – the largest event ever recorded in the area – we unveil a 3D geometry for the AIFS. We report evidence of left-lateral strike-slip displacement, characterise the fault segmentation and demonstrate that AIFS is the source of the 2016 events. The occurrence of the $M_w$ 6.4 earthquake together with historical and instrumental events supports that the AIFS is currently growing through propagation and linkage of its segments. Thus, the AIFS provides a unique model of the inception and growth of a young plate boundary fault system.

[1] Barcelona-CSI, Institut de Ciències del Mar, ICM-CSIC, 08003 Barcelona, Spain. [2] GEOMAR Helmholtz Centre for Ocean Research, 24148 Kiel, Germany. [3] GRD, Scripps Institution of Oceanography – UCSD, CA92093 La Jolla, San Diego, USA. [4] Irish Centre for Research in Applied Geosciences (iCRAG), University College of Dublin, School of Earth Sciences, Belfield, Dublin 4, Ireland. [5] Institute of Earth Sciences Jaume Almera, ICTJA-CSIC, 08028 Barcelona, Spain. [6] Institut de Physique du Globe de Paris, UMR7154 CNRS, 75005 Paris, France. [7] National Oceanography Centre, Waterfront Campus, Southampton SO14 3ZH, UK. [8] Unitat de Tecnologia Marina, UTM-CSIC, 08003 Barcelona, Spain. [9] Sorbonne Universités, UPMC Univ Paris 06, CNRS - ISTEP, 75252 Paris, France. [10] Institució Catalana de Recerca i Estudis Avançats (ICREA), 08010 Barcelona, Spain. Correspondence and requests for materials should be addressed to E.Gàc. (email: egracia@icm.csic.es)

The Alboran Sea is a Neogene basin in the westernmost Mediterranean Sea, located between the Iberia and Nubia plates (Fig. 1). Miocene deformation related to roll-back of the Tethys oceanic lithosphere[1] was followed by a compressional regime, which lasted from the Pliocene until today[2,3], and included the development of strike-slip and thrust faults[4,5] (Fig. 1). Present-day crustal deformation is driven by the fault systems within the overall plate tectonic setting of NW-SE to NNW-SSE trending convergence (4.5–5.6 mm/yr) between the Nubian and Eurasian plates[6] (Fig. 1). Seismicity in the study area is characterised by earthquakes of small to moderate magnitude[7]. Large historical and instrumental earthquakes have occurred in the region, such as the 1804 and 1910[8] Adra earthquakes (MSK Intensity VIII–X), and the $M_w$ 6.0, 26 May 1994[9,10] and the $M_w$ 6.3, 24 February 2004[11] Al-Hoceima earthquakes (Fig. 2a). This last event caused 629 fatalities and left 15,600 homeless[12], making it the most catastrophic earthquake in the region during the last century. On 25 January 2016, a $M_w$ 6.4 earthquake (white star in Fig. 1) hit the area offshore the city of Al-Hoceima on the Moroccan coast[13,14]. This is the largest event recorded in the Alboran Sea. The earthquake caused one casualty in Al-Hoceima and 30 injured in Melilla. Damages were reported in several coastal cities of northern Morocco and southern Spain, where the event was strongly felt (i.e. Intensity V (EMS-98) in Malaga)[15].

Besides the intermediate (>100-km-depth) seismicity in the West Alboran Basin related to the east-dipping Rif-Gibraltar-Betics slab[16,17], an ~80-km-wide NE-SW trending seismic zone extends for ~500-km-long[18] and runs along the so-called Trans-Alboran Shear Zone (TASZ)[19]. The TASZ is traditionally interpreted as a complex belt of deformation that crosscuts the Alboran Sea and its two margins, connecting the Rif (North Africa) to the Eastern Betic Shear Zone (SE Iberian Peninsula)[18,19]. Only a few works have proposed that the TASZ may play the role of a plate boundary across the Alboran Sea, traversing the Nubia-Eurasia plates in the westernmost Mediterranean[20–22]. Its associated seismicity is characterised by left-lateral strike-slip focal mechanisms with few normal and thrust fault plane solutions[18] (Figs. 1 and 2). A recent work that combines geological, geodetic and 3D numerical modelling[17] demonstrates that crustal deformation in the Alboran Sea, induced by NNE-directed dragging of the RGB slab by the Nubia plate in the past 8 Myr, is still active. The slab dragging is resisted by the mantle and this resistance translates into further crustal deformation at the surface[17]. Such recent deformation has been documented, for example, along the Yusuf Fault System, Carboneras Fault System and especially along the AIFS and associated structures of the Rif and the Eastern Betics Shear Zone (Figs. 1 and 2b), which comprise the main fault systems of the TASZ[19] (Figs. 1 and 2).

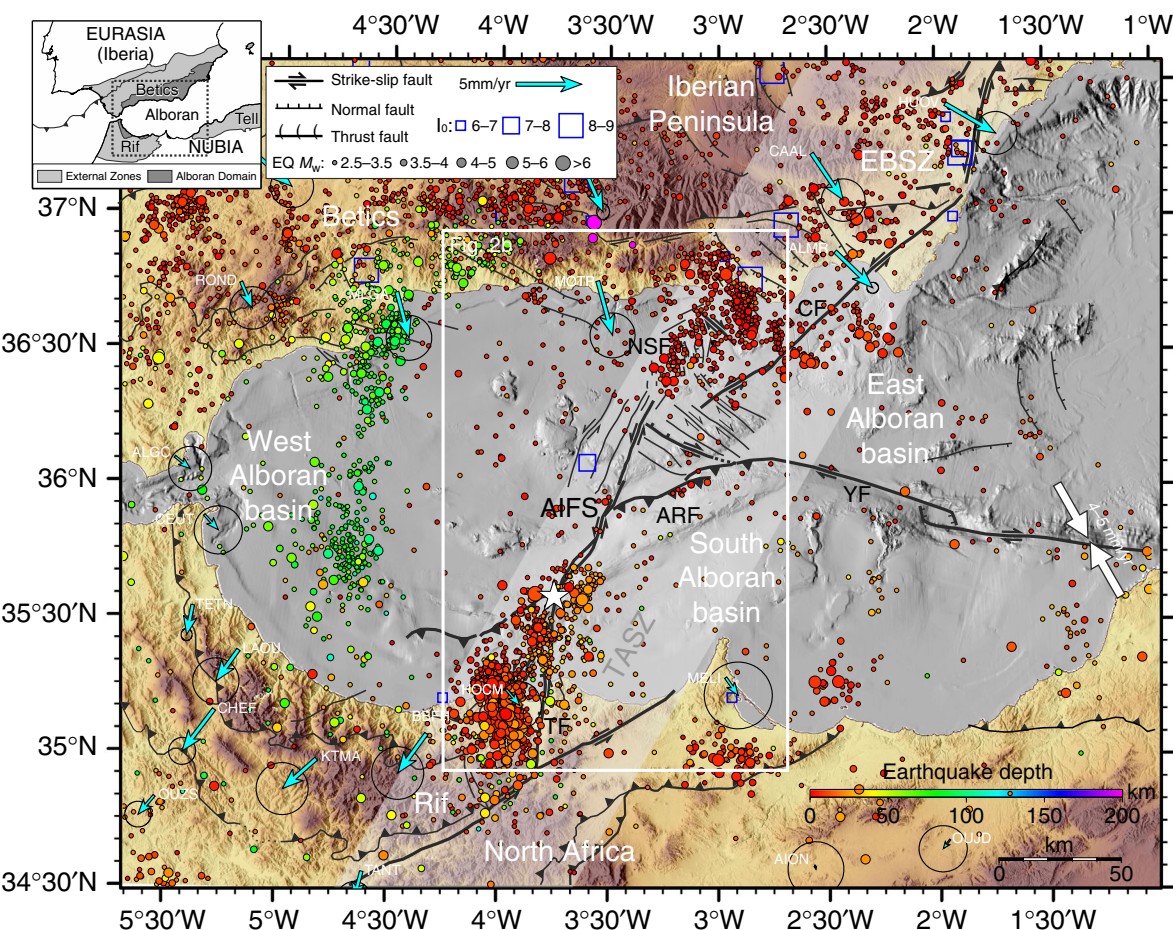

**Fig. 1** Tectonic setting, seismicity and GPS velocities in the Alboran Sea region. Topography and bathymetry of the Alboran Sea. Historical (1400–1960) and instrumental (1980–2015) seismicity previous to the 2016 event are colour-coded according to hypocentre depths. The GPS velocity field is drawn with respect to Nubia[24] (blue arrows), and Eurasia-Nubia relative motion[6] (white arrows). The main faults located on the map are: AIFS: Al-Idrissi Fault System; ARFS: Alboran Ridge Fault System; CFS: Carboneras Fault System; NSF: North-South Faults; EBSZ: Eastern Betic Shear Zone; TASZ: Trans-Alboran Shear Zone; TF: Trougout Fault. HOCM: Al-Hoceima; YFS: Yusuf Fault System; MELI: Melilla. Upper left: Configuration of the Eurasia and Nubia Plates in the western Mediterranean Sea

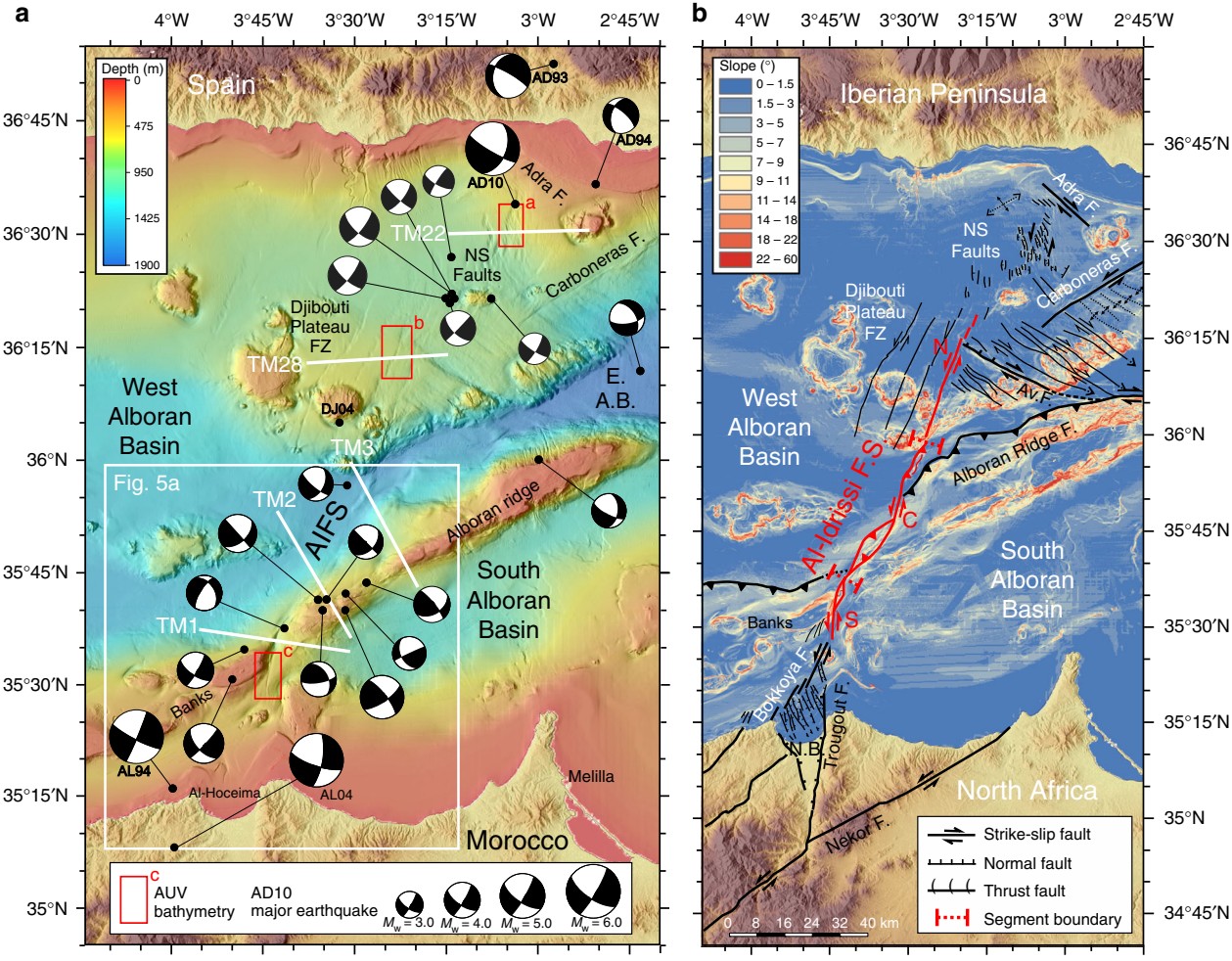

**Fig. 2** Shipboard bathymetry, earthquake focal mechanisms and active faults. **a** Shipboard bathymetry and moment-tensor solutions for earthquakes that occurred in the area before the 2016 events. Major earthquakes of $M_w > 4.8$ are noted: DJ04: Djibouti plateau 1804[15]; AD10: Adra 1910[25]; AD93–94: Adra series 1993–1994[15]; AL94: Al-Hoceima 1994[9,10]; AL04: Al-Hoceima 2004[11]. White lines locate the MCS profiles (Fig. 4). Red boxes locate Fig. 3a–c. EAB: East Alboran Basin. **b** The Al-Idrissi Fault System (in red) and related tectonic structures are overlaid on a slope map. AIFS North (N), Central (C) and South (S) segments are located on the map

Here we present a new and comprehensive geological and geophysical dataset of the entire AIFS. We adopted a multi-scale approach, including detailed morphological analysis of shipboard multibeam bathymetry and near-bottom bathymetry obtained with Autonomous Underwater Vehicles (AUVs), and inter-pretation of deep penetration multichannel-seismic (MCS) data (Figs. 2a and 3). Combining these data with the analyses of the $M_w$ 6.4 earthquake in 2016 provides us with a unique opportunity to explore the role of seismic deformation in the fault propagation and growth of a continental fault system.

## Results

**Seafloor expression of the fault system.** Although the AIFS shows a subdued topography, it may still represent the longest active tectonic structure in the region. The AIFS is a Plio-Quaternary[4,23] structure that offsets the largest bathymetric relief of the basin, separating the prominent thrust of the Alboran Ridge Fault System, of Early Pliocene age[4,23], from several banks to the SW (Fig. 2a, b). The AIFS is a left-lateral fault system trending NNE-SSW. The fault system is about ~ 100-km-long, with a width varying from 1-to-4.8-km-wide. Considering that the AIFS is the major fault structure in a large region, we infer that it accommodates most of the total rate of 3.8 mm/yr[24]

(Figs. 1 and 2b). The AIFS runs from the Djibouti Plateau in the north, where the historical 1804 earthquake occurred (DJ04, MSK Intensity VIII)[8](Fig. 2a), to the Nekor Basin (Moroccan margin) in the south (Fig. 2b). Towards the North, the AIFS connects to a parallel structure, a wide shear-zone defined as the NS Faults system (NSF)[5], located near where the destructive 1910 Adra earthquake (AD10, estimated $M_w$ ~6.1)[25] and the 1993–1994 seismic crises (AD93–94) occurred[7,15] (Fig. 2a). To the south, the AIFS links to the Trougout and Bokkoya faults (Nekor Basin), the last of which is related to the source of the Al-Hoceima 1994 earthquake[8–10] (Fig. 2a, b).

From 2006 to 2016, a series of shipboard bathymetry campaigns were carried out along the central part of the Alboran Sea[3–5,26] (gridded at 20 m/pixel) to complete the mapping of the AIFS and related structures (Fig. 2a, b). In 2015, a few months before the 2016 earthquake, we collected high-resolution near-bottom bathymetry data (at 1-metre resolution) of three sections of the AIFS segments (Fig. 3a–c) using multibeam sonars mounted on two AUVs (see Methods). Multi-scale acoustic mapping techniques, such as swath-bathymetry allow identifying the geomorphological expression of active faults, such as seafloor ruptures, fault scarps and fault traces[27,28]. The AUV bathymetric data clearly highlight the surface expression of the AIFS, whose

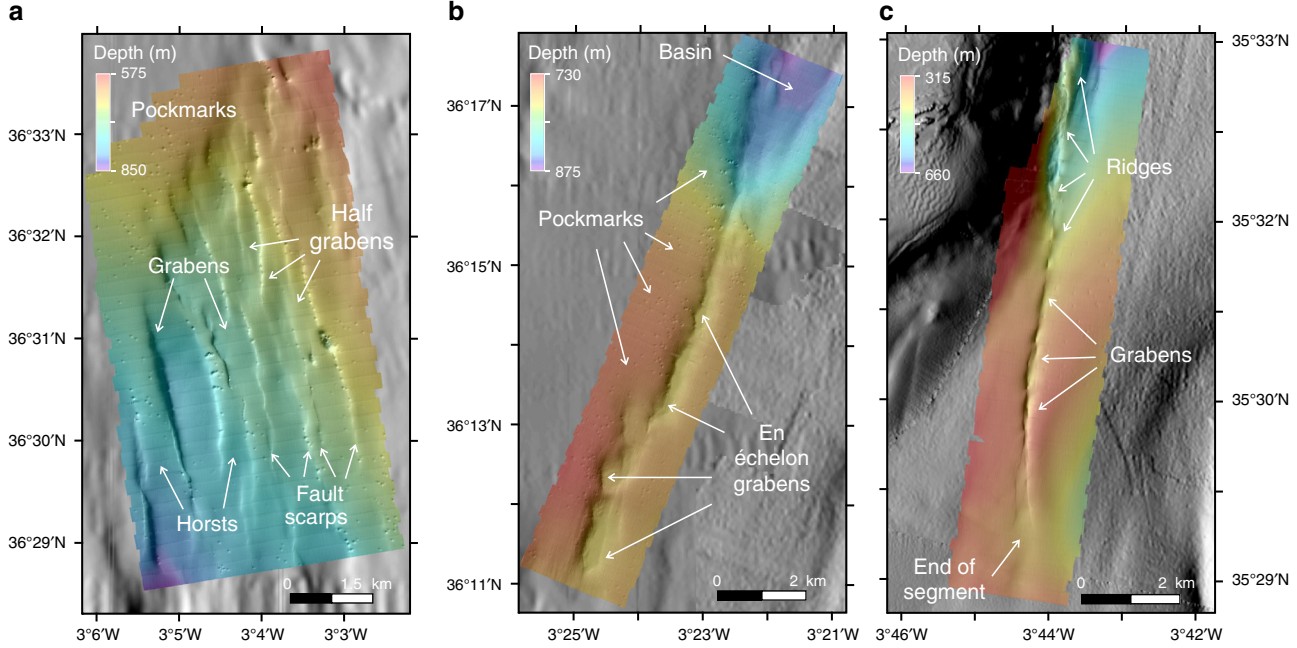

**Fig. 3** AUV near-bottom bathymetries. They are gridded at 1-m resolution corresponding to the NS faults (**a**), the North AIFS (**b**) and the South AIFS (**c**) segments. AUV bathymetric maps were acquired using the Autonomous Underwater Vehicles (AUVs) IdefX and AsterX (France)

trace reaches and offsets the seafloor, indicating recent fault activity. There are abundant pockmarks, only visible in the near-bottom bathymetry near the AIFS traces, suggesting past and possibly on-going fluid flow[29] circulation activity (Fig. 3a–c).

According to the fault trend, geometry and timing of activity, the AIFS can be divided into three main segments: north, central and south. Related nearby structures, such as the NSF and the Bokkoya and Trougout faults are also considered part of the AIFS (Fig. 2b). The NSF is a ~20-km-long, 5-km-wide left-lateral shear zone composed of a succession of closely spaced N160 trending en echelon elongated troughs (Figs. 2a and 3a). South of it, the North AIFS segment is 34-km-long, it trends N018 and is of Quaternary age[4,23]. This segment runs across the Djibouti Plateau FZ composed by a magmatic arc crust and magmatic intrusions[30,31]. It is cut by four closely spaced sets of parallel fault arrays (Fig. 2b). The North AIFS (easternmost array) consists of a succession of single N10-N20 trending en echelon elongated troughs (Figs. 2a and 3b). Southwards, a 2-km-wide left-stepping offset in the fault trace around 36°N/3°28′W marks the boundary between the North and Central AIFS (Fig. 2b). The Central AIFS segment trends N031, is 50-km-long and of Late Pliocene to Quaternary age[4,23]. It is the longest and most mature segment of the AIFS, and it includes a principal displacement zone (PDZ) that is ~26-km-long, where most horizontal displacement is accounted for. Two NE–SW trending compressional zones can be observed in the slope map (Fig. 2b). A change in strike of the fault trace defines the intersection between the Central and South AIFS, which correspond to a releasing bend (Fig. 2b). The South AIFS segment trends N007, it is 16-km-long, and of Quaternary age[4,23]. The active fault trace defines an elongated sigmoid, which is composed of a succession of en echelon narrow grabens that evolve to the north into a succession of small pressure ridges, which do not appear to be fully connected. Towards its southern end, the AIFS loses its surface expression and splits into two branches (Figs. 2a and 3c). Further south, along the Moroccan Margin, strain is transferred towards the Bokkoya and Trougout faults, which are connected through an

intricate network of normal faults accommodating deformation in the Nekor pull-apart basin[26] (Fig. 2b).

**Subsurface structure of the fault system**. To examine the crustal structure of the AIFS segments, we selected five MCS profiles across the TASZ obtained during fall 2011 (Figs. 2a and 4). The seismic profiles are displayed in two-way travel time (first 6s TWTT), except for profile TM3, which is in depth (first 5 km) (see Methods and Supplementary Fig. 1), showing in detail the upper crustal structures. Subsurface images show the tectonic architecture of the AIFS, a sub-vertical, left-lateral strike-slip fault that roots into the basement. Deformation cuts through the most recent sediments up to the seafloor (Fig. 4). In Supplementary Fig. 2 we provide an example of a lithospheric-scale profile across the AIFS (12s TWTT record length) that shows two different crustal domains: The West Alboran continental crust (about ~8s TWTT thick) and the North African continental crust (about ~10.5s TWTT thick)[32].

The age of deformation is derived from the seismo-stratigraphic interpretation combined with scientific and commercial wells available in the Alboran Basin[4,23,31,33]. We assume that if a given stratum displays a roughly constant thickness across the fault zone, then that stratum must have been deposited prior to the initiation of a particular fault segment. For example, in profiles TM2 and TM3 (Central AIFS segment), deformation starts in the Early Pliocene (Unit IIc-d), while in profiles TM1 (South AIFS segment) and TM28 (North AIFS segment), deformation starts post unit IIa-b (Early Pleistocene to Holocene in the Quaternary). Our interpretation fits with older data and more recently acquired MCS profiles in the Alboran Basin[4,23,31,33]. Hence, this allows proposing a new seismo-stratigraphic model[34] (Fig. 4). Above the basement, the following units are identified, Ia-b: Late Pleistocene-Holocene (Quaternary), IIa: Early Pleistocene (Quaternary), IIb: Late Pliocene, IIc-d: Early Pliocene, III: Messinian (Late Miocene), IV: Late Tortonian (Late Miocene), V: Late Serravalian-Early Tortonian (Middle Miocene-Late Miocene); VI: Langhian-Serravalian (Middle Miocene), and

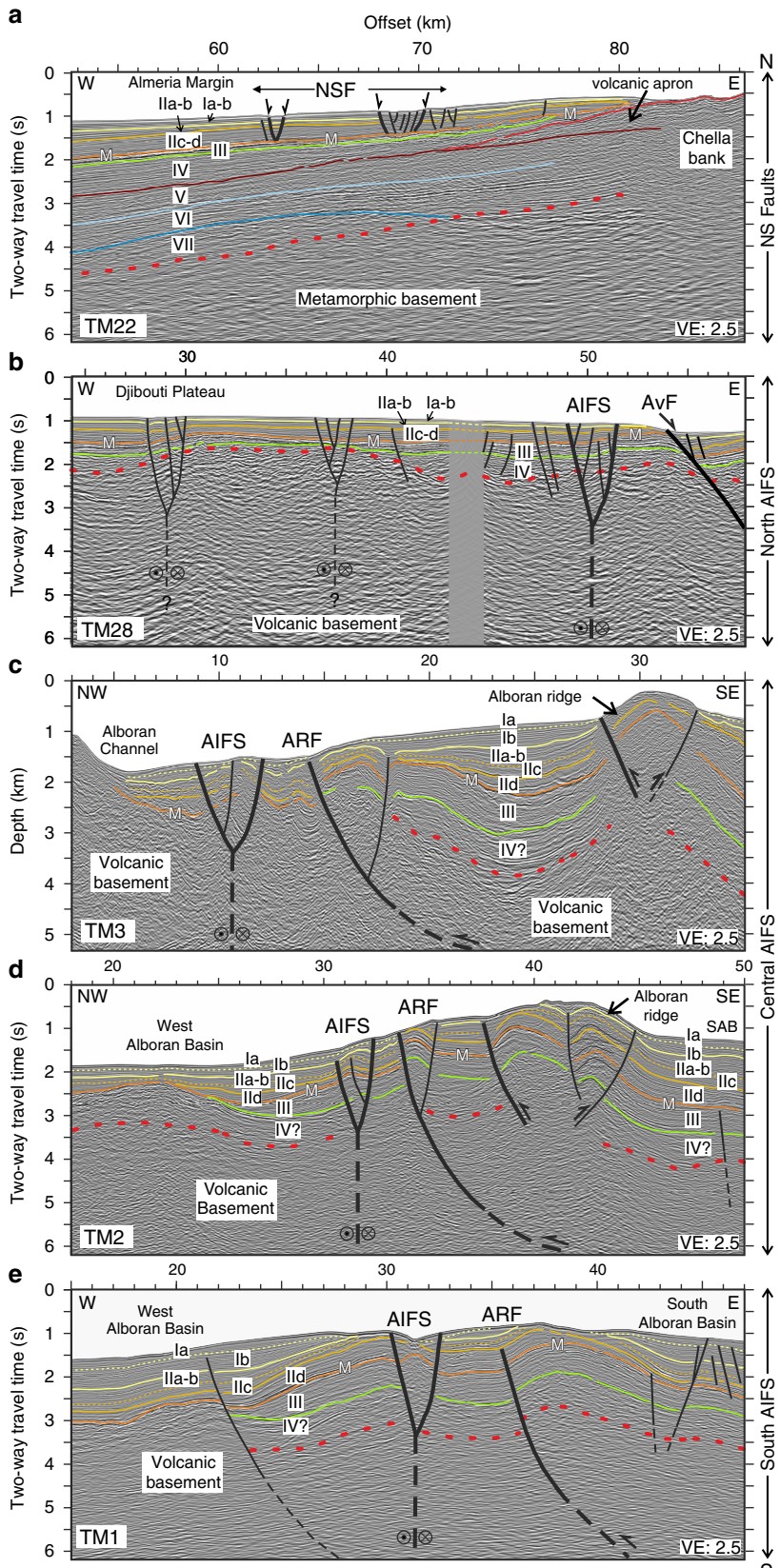

**Fig. 4** Interpreted seismic reflection profiles across the AIFS. From north (top) to south (bottom), multichannel seismic profiles illustrate the geometry and tectonic pattern of the AIFS. The five profiles are located in Fig. 2a. **a** NSF (profile TM22); **b** North AIFS segment (profile TM28); **c** Central AIFS segment - pressure ridge (profile TM3); **d** Central AIFS segment - restraining bend (profile TM2); and **e** South AIFS segment - releasing bend (profile TM1). All profiles are plotted in two-way travel time (s) except for TM3, which is in depth (km). Ages of seismo-stratigraphic units are detailed in the text. AIFS: Al-Idrissi Fault System, ARFS: Alboran Ridge Fault System, AvF: Averroes Fault, M: Top Messinian horizon, SAB: South Alboran Basin

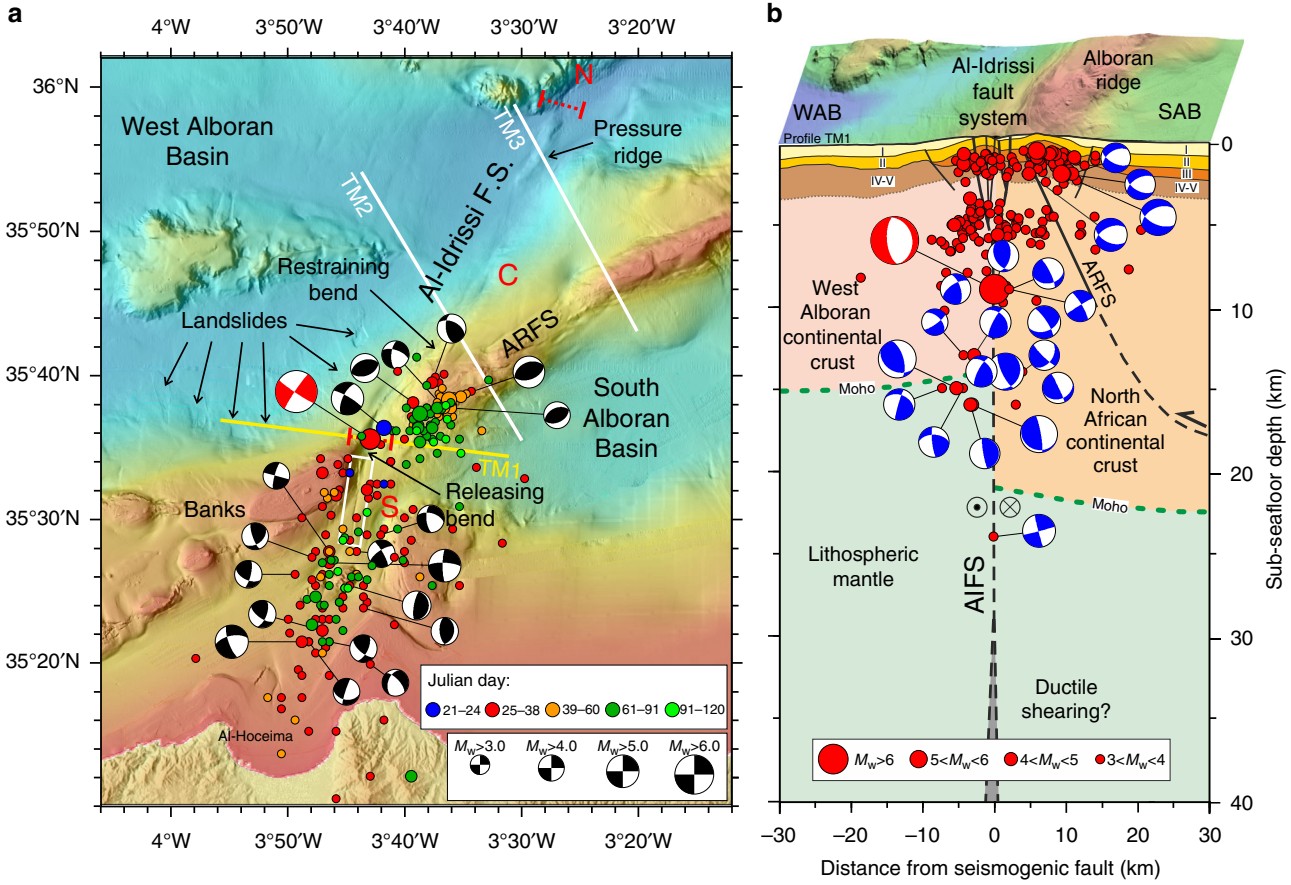

**Fig. 5** Aftershock analysis, focal mechanisms and conceptual AIFS section. **a** Bathymetric map showing relocated aftershocks over time. Focal mechanisms of the 2016 main shock (red), foreshock and aftershocks (black) are included. Profile TM1 is depicted in yellow. The North (N), Central (C) and South (S) segments are located on the map. **b** Conceptual lithospheric section across profile TM1 with focal mechanism of the mainshock (red) and aftershocks (blue). Focal mechanisms have been projected onto the vertical plane of the focal sphere. Earthquake hypocentres (red dots) of $M_w$ 3–6.4 are projected within ±30 km of profile TM1. Faults and structure of the Moroccan Margin are shown. ARFS: Alboran Ridge Fault System, WAB: West Alboran Basin, SAB: South Alboran Basin

VII: Burdigalian (Early Miocene). The metamorphic basement is of Late Oligocene-Early Miocene age[31], and the volcanic basement is of Late Serravalian-Tortonian age[35,36].

The seismic profiles displayed in Fig. 4 supports evidence for the inception of fault activity. To the north, the NSF is composed by half-grabens and horst-and-graben structures (Fig. 4a), which are active and consistent with the present-day extensional strain pattern of this area. The North AIFS shows a sub-vertical, left-lateral transtensional strike-slip fault (Fig. 4b). The Central AIFS segment shows local folding and reverse faulting deformation consisting, from north to south, of a 2.5-km-wide and 10-km-long pressure ridge (Fig. 4c), and a 4.8-km-wide and ~18-km-long restraining (compressional) bend with a positive flower structure (Fig. 4d). Narrow folds and sub-vertical faults extending down to at least 5 km depth are observed in the seismic profiles (Fig. 4c). To the south, across a 2-km-wide and 3.7-km-long releasing bend, the seismic image shows wide folding over the AIFS, which converges at depth to form a flower structure. The occurrence of growth-strata in the Late Pliocene to Quaternary units (IIa-b to Ib) west of the AIFS is consistent with the ongoing fault activity (Fig. 4e).

**The 2016 $M_w$ 6.4 earthquake and relocated seismicity.** The epicentre of the $M_w$ 6.4 earthquake on 25 January 2016 was located in the Alboran Sea[13–15], about 42 km north of the city of

Al-Hoceima (Morocco; Figs. 1 and 5a). The mainshock was preceded on 21 January by a foreshock of magnitude $M_w$ 5.1 located in the same epicentral area. The mainshock was also followed by an extensive aftershock sequence of >2350 events (i.e. from the 25 January until 13 May 2016)[15], included 197 events of magnitude $M_w \geq 3$ (Fig. 5a).

Using a local lithospheric velocity model[37] we relocated the mainshock as well as the aftershocks (see Methods). For the mainshock, we located the epicentre at 35.59°N and 3.72°W, which corresponds to a transtensional releasing bend between the Central and South AIFS segments (Fig. 5a). The moment-tensor waveform-inversion (see Methods) yields a preferred depth of 10 km, and left-lateral strike-slip focal mechanisms with a preferred nodal plane of 214°/85°/5° (strike/dip/rake). The strike is consistent with the azimuth of the AIFS (Fig. 5a and Supplementary Fig. 3). The slip propagated northward for <16 km (Supplementary Fig. 4) with a maximum coseismic slip of about 1 m, which might have ruptured the seafloor south of the epicentral area (Supplementary Figs. 4 and 5). Aftershocks were distributed along the southernmost part of the Central AIFS segment and the whole South AIFS segment. A significant number of aftershocks were also located at the western tip of the Alboran Ridge Fault System (Fig. 5a).

The relocated aftershocks for the first four weeks (days 21–53) roughly outline the mainshock fault trace (Fig. 5a, Supplementary Fig. 6). Their focal mechanisms (i.e., the

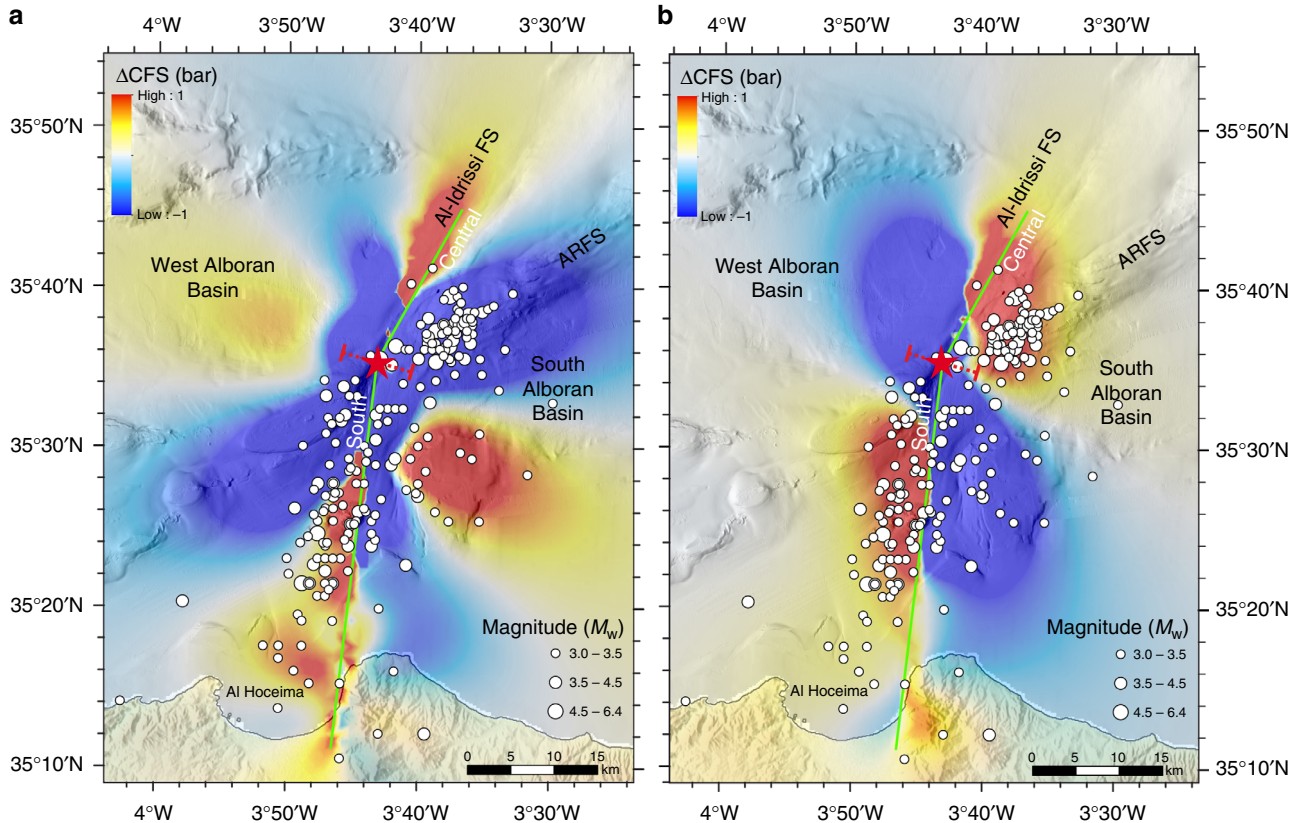

**Fig. 6** Coulomb stress transfer modelling. **a** Calculated Coulomb failure stress change[41, 42] (ΔCFS) at 10 km depth on receiver faults striking 210°, dipping 90° and with a rake of 5°. **b** Calculated Coulomb failure stress change (ΔCFS) at 10 km depth on receiver faults striking 070°, dipping 45° and with a rake of 85°. The source fault (light green line) mimics the coseismic slip determined from inversion of teleseismic waveforms (Supplementary Fig. 3), and corresponds to a vertical left-lateral strike-slip fault (rake 5°) that bends in the epicentral area. The South AIFS rupture strikes 007°N and extends for 45 km, while the Central AIFS rupture strikes 031°N and extends for 20 km. The boundary between the South and Central AIFS segments is depicted by a dashed red line

minimum magnitude is $M_w > 3.6$) and centroid depth (regional waveform inversion) mimicked the mainshock, with preferred fault planes striking parallel to the AIFS (Fig. 5a, b, Supplementary Table 1). Later aftershocks (days 54–120) appeared near the southern and northern terminations of the rupture area and further east (Fig. 5a, Supplementary Fig. 6), supporting stress transfer into adjacent faults such as Bokkoya and Trougout Faults (Fig. 2b) and the Alboran Ridge Fault System thrust. Consequently, strain partitioning between these two types of tectonic structures (i.e., the strike-slip dominates in the South AIFS segment and thrust in the southern part of the Alboran Ridge Fault System) accommodates different components of the total motion[38]. Regional centroid moment-tensor solutions (RCMTs) were obtained for the largest foreshocks and aftershocks of the earthquake sequence using a full waveform inversion (see Methods for more details and Supplementary Fig. 7). Moment-tensor solutions for the Alboran Ridge Fault System aftershocks are compatible with a NE-SW-oriented thrust, demonstrating that slip may occur on this fault (Fig. 5, Supplementary Table 1).

The mainshock and aftershocks were relocated using a local lithospheric velocity model adjusted for the offshore domains (Supplementary Fig. 8a). We corrected the 1D velocity model for effects caused by 3D propagation in a heterogeneous setting by introducing station correction terms. The station terms compensate differences in the velocity structure caused by structural heterogeneity between the onshore and offshore domains, and hence provide an approximation of the 3D velocity structure. The mainshock and aftershocks fall on the trace of the AIFS, which is

similar to the relocations obtained with a regional 3D velocity model[13] (Supplementary Fig. 8b). Alternative interpretations used the relocations of a 1D Iberian velocity model[15,39] (Supplementary Fig. 8c), in which the mainshock and aftershocks are located 15 km west of the fault trace, inferring that the main event was generated by an undetected fault[39]. This fault is questionable because generating a $M_w$ 6.4 earthquake would require a rupture of at least 30-km-long fault, as supported by the source inversion (Supplementary Fig. 4). This implies a cumulative fault offset of several hundred metres[40], which should have been detected with seismic images.

In the conceptual section across the profile TM1 (Fig. 5b), the AIFS delineates a boundary between two different crustal domains: to the east, the South Alboran Basin is mainly formed by the North African continental crust[3,34], and to the west, the West Alboran Basin is characterised by a thin continental crust[3,34]. The presence of a relatively thin continental crust (~15–23 km) on both sides of the AIFS, together with high heat-flow[37], restricts the depth of the seismogenic zone (Fig. 5b), supporting a rupture at <15–20 km depth.

To illustrate how moderate to large earthquakes might exert control on the distribution of the aftershocks and could possibly trigger large earthquakes along the AIFS and nearby faults, we modelled the change in Coulomb failure stress[41,42] (ΔCFS; see methods). The source fault (green line in Fig. 6a, b) mimics the coseismic slip determined from the inversion of teleseismic waveforms (Supplementary Fig. 4), with similar geometry to those described for the South and Central AIFS segments. Strike-

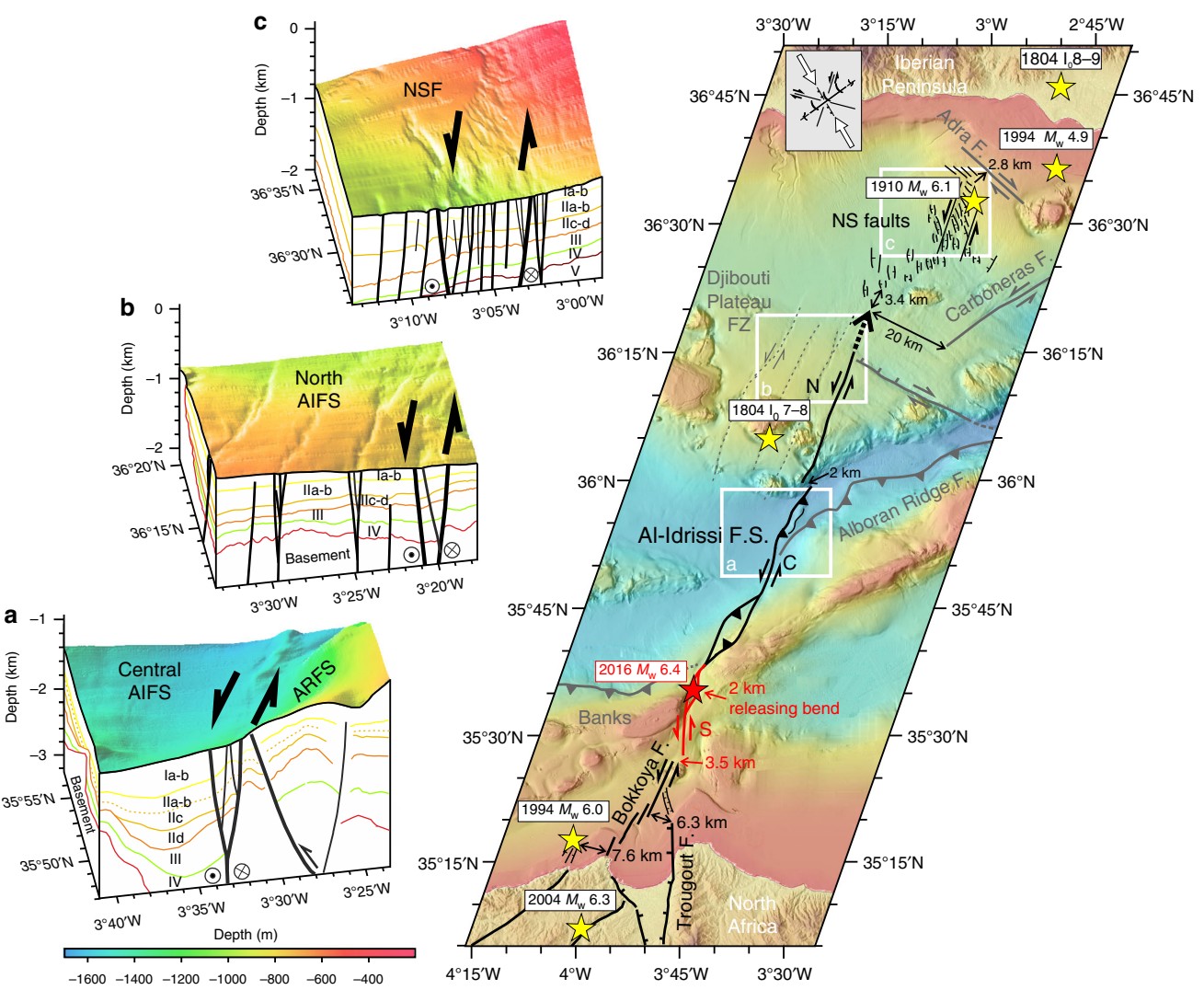

**Fig. 7** Evolutionary stages of AIFS and conceptual model of fault growth. Traces of active fault segments that likely ruptured during the 2016 earthquake are shown as red lines. Red star: mainshock; Yellow stars: epicentres of previous large magnitude earthquakes: 1994[9, 10] and 2004 events in north Morocco[11], 1804 (Djibouti $I_0$7–8 and Adra $I_0$8–9)[15], 1910 (NSF)[25] and 1994 (Adra)[15] events in south Spain. Red and black arrows point to fault gaps and steps measured perpendicular to the fault strike. Offsets are labelled. The boundaries between the AIFS segments are depicted by dashed red lines. **a–c** are three-dimensional views representing snapshots of the AIFS evolutionary stages

slip and thrust receiver faults are defined, respectively, by strike/dip/rake of 210°/90°/5° and 070°/45°/85° (Fig. 6). The comparison between the distribution of the increase stress lobes and the location of the aftershocks shows a good spatial correlation (Fig. 6). However, it is also noticeable that the increase in stress at the southern tip of the Central AIFS segment is not associated with the occurrence of aftershocks. This may suggest that the 2016 earthquake increased the level of stress along the southern part of the Central segment, which was not released by an aftershock, bringing the Central AIFS segment closer to failure (Fig. 6a).

## Discussion

Continental earthquakes usually rupture active fault sections[43,44] that are bounded by discontinuities such as bends, step-overs, gaps or branches[45,46]. These discontinuities have been recognised as favourable for initiating and stopping earthquakes[47], such as the 2016 $M_w$ 6.4 event, which started in a 2-km-wide releasing bend (Fig. 5a). The en echelon structural pattern of the South

AIFS segment probably controlled the 2016 $M_w$ 6.4 event rupture: short segmented faults generate small displacements and are thus often associated with small magnitude earthquakes[48], as indicated by low magnitude aftershocks located along the South AIFS. Over time, however, accumulation of seismic slip might lead to simpler fault geometry, and eventually longer strike-slip fault zones[45] with the potential for larger magnitude earthquakes. This might already be the case for the longest segment of the AIFS, the Central segment, which shows a well-defined PDZ (Figs. 2b and 7a). In contrast, the Northern and Southern AIFS segments, which are younger[4,23,34], show more discontinuous fault traces and appear to be in an earlier stage of fault development[48] (Fig. 7b, c).

Fault growth, subsequent lateral propagation and fault linkage[48,49] between the Central AIFS and North AIFS segments may be possible, as the transfer of slip between the two fault steps would occur over a short distance (2 km; Fig. 7), which is smaller than the established empirical limit for step-over jumps for strike-slip faults (i.e. generally ~4 km[45] to ≥ 6 km[46]). The North AIFS shows shallow en echelon grabens, which may eventually link by

lateral growth and merge at depth (Fig. 7b). Thus, on-going and future linkage of the North and Central AIFS segments through the entire brittle crust, may generate longer faults and increase the seismic potential of the overall fault system[50].

The Southern AIFS, and associated structures further south along the Moroccan Margin (i.e. Bokkoya and Trougout faults), released elastic strain energy during the 2016 seismic crisis and previously in 1994 and 2004. Along the northern margin of the Alboran Sea, a series of earthquakes occurred during the last 200 years, including in two historical events in year 1804 ($I_0$ 8–9 and $I_0$ 7–8) and the instrumental 1910 and 1994 events (Fig. 7c). However, no significant earthquake has been reported during the historical and instrumental periods[15] along the entire Central AIFS segment (Fig. 7a) and most of the North AIFS segment (Fig. 7b), with the exception of the 1804 ($I_0$ 7–8)[21] earthquake located on a parallel trace (Fig. 7). This observation may indicate that these segments are either locked or possibly creeping[51]. Hence, the AIFS may have the potential to generate larger events if earthquakes manage to propagate across fault step-overs and generate multiple-segment ruptures[49], as it has been proposed for other fault systems, such as the southern San Andreas Fault System (SAFS) in California[52] or the Dead Sea Fault System (DSFS)[53] in the eastern Mediterranean Basin, for example.

The AIFS is a unique example of a young continental fault system that is currently in an incipient stage. It is growing and, in the course of time, could develop into a large-scale continental plate-boundary fault along the Trans-Alboran Shear Zone (Fig. 7), similar to the North Anatolian Fault System (NAFS)[54,55] or the SAFS[56]. Although the AIFS is accommodating a slip-rate of ~3.8 mm/yr[24], an order of magnitude lower that the NAFS or SAFS, and comparable to the DSFS[53] slip-rate, all these systems form major lithosphere cutting faults between tectonic plates, extending for more than 1000 km, and generating large magnitude earthquakes.

Earthquake hazard assessment models are based on the potential length of seismic ruptures and whether rupture might stop or not at fault-segment boundaries, to determine the difference between a moderate and a potentially devastating earthquake[45,57]. Regarding the seismic potential of the AIFS, using classical scaling laws that relate magnitude to rupture length[57], we can envision several scenarios depending on the potential length of fault activated. In a worst-case scenario, considering a rupture that would include the segment ruptured in 2016, the South AIFS segment, together with the North and Central AIFS segments, and the faults located at short distances from the endpoints of the AIFS segments (i.e. such as the 25-km-long Bokkoya fault, located 3.8 km to the SW of the South AIFS segment, and the 35-km-long left-lateral NSF located ~3.4 km to the NE of the North AIFS segment), eventually, it may result in a maximum rupture of 160 km. This may yield an earthquake of maximum magnitude $M_w$ 7.5 ± 0.2 to 7.6 ± 0.3 across the entire TASZ from the Moroccan to the Spanish margins (Fig. 7).

A sequence of historical (AD1804 and AD1910)[8] and instrumental (1994, 2004 and 2016)[9–11] earthquakes with estimated magnitudes ranging from $M_w$ 5.9–6.4, has hit the Alboran Sea region in northern Morocco and southern Spain in the last 200 years (Fig. 7). Given the low awareness and preparedness for seismic and tsunami hazards[58,59] in the region, a major earthquake may eventually cause severe damage along the highly populated coastal zones of the Alboran Sea. Therefore, large events should be considered in future seismic and tsunami hazard assessments and mitigation plans. The recent deformation that we now observe along the AIFS, from North Morocco to the Eastern Betic Shear Zone that we refer to as the TASZ[15] (Figs. 1 and 7), may represent a plate boundary that will eventually develop into a mature, large-scale continental plate-boundary fault zone[17].

## Methods

**Multibeam bathymetry**. Multibeam shipboard bathymetry was acquired during the 2006 IEO and IMPULS, 2010 EVENT-DEEP, 2011 TOPOMED-GASSIS, 2012 SARAS and MARLBORO-2, 2015 SHAKE, and 2016 IDRISSI cruises. Hull-mounted multibeam data along the AIFS were acquired with a 1°×1° beam width Atlas Hydrosweep DS multibeam echosounder (R/V Sarmiento de Gamboa) and were processed with the CARIS HIPS&SIPS 9.0 software and gridded at 20 m resolution. For the whole Alboran Sea we used the IEO 25 m multibeam compilation[60]. Ultra-high-resolution, near-bottom bathymetry data were acquired with a Simrad EM2040 multibeam echosounder installed on the Autonomous Underwater Vehicles (AUVs) AsterX and IdefX from IFREMER (France) during the SHAKE cruise onboard the R/V Sarmiento de Gamboa in May 2015. The AUV surveys were conducted at ~ 70 m above the seafloor, in the North, Central and South AIFS segments covering areas of 43, 40 and 32 km², respectively. The AUV inertial navigation was corrected using the ultra-short baseline (USBL) acoustic navigation as a reference. The navigation of the AUV was ultimately post-processed and corrected with the CARAIBES 4.3 software. Multibeam bathymetry was processed with the CARIS HIPS&SIPS 9.0 software and gridded at 1 m cell size.

**Seismic reflection**. The multichannel seismic (MCS) profiles used in this work were acquired during the 2011 TOPOMED-GASSIS cruise onboard the Spanish R/V Sarmiento de Gamboa. During the TOPOMED-GASSIS cruise, multichannel seismic data were acquired using a 50.15 l (3060 ci) airgun source composed by 8 G-GUN II guns deployed at 7.5–9 m depth working at 2000 to 2500 psi, in a single and cluster distribution geometry of five gun positions: 380 × 2, 520, 250 × 2, 520 and 380 × 2 (c.i.). Seismic signals were acquired with a 5100–6000-m-long active section of a Sentinel Sercel streamer composed by 408 to 480 active sections (12.5 m channel interval) at a depth of 10 m. Profiles TM1 and TM2 were fired at 2500 psi every ~ 30 m, while profiles TM3, TM22 and TM28 were fired at 2000 psi every ~ 40 m. Data were recorded at a sample rate of 2 ms and a record length of 12–14 s, except for TM28, which was recorded at a record length of 19 s. The MCS profiles were processed using Globe Claritas seismic processing software (http://www.globeclaritas.com). The processing flow was designed in order to obtain the best image in both the shallower and deeper parts of the profile[3,34]. The processing sequence included the insertion of the geometry accounting for streamer feathering; a minimum phase conversion; spherical divergence correction; predictive deconvolution in the Tau-P domain (to eliminate the bubble and short period multiple reverberations); surface consistent deconvolution; Surface Related Multiple Elimination (SRME) demultiple; normal-move-out correction based on semblance velocity analysis; Radon filter demultiple; Dip Move-Out (DMO) correction; a zero-phase conversion and a time migration. The final image had a time and spatial variant filter to remove the incoherent noise, and an amplitude correction was applied[3,34]. Furthermore, we performed a Pre-Stack Depth Migration (PSDM) of profile TM3 with the software Echos by Paradigm (http://www.pdgm.com/products/echos/) to obtain the real geometry of the structures in depth[34]. We applied the same processing flow to the shot gathers (till the DMO correction) and obtained the depth velocity model needed for the migration through an accuracy velocity analysis based on the residual analysis, first in the time domain and second in depth. Finally, we exported the time and depth migrated sections in SEG-Y format.

**Teleseismic waveform inversion of the mainshock**. To study the mainshock of the crisis on 25 January 2016, we retrieved data from the Incorporated Research Institutions for Seismology (IRIS) data management system (DMS) for waveform inversion. Waveforms were first converted to displacement by removing the instrument response in the frequency range lower than 1 Hz. Second, we used an iterative least-squares inversion[61] of azimuthally distributed seismic P and SH body-wave signals from stations at distances of about 30° to 90° to determine the rupture mechanism, depth and an initial source time function. Waveforms were corrected for instrument responses to obtain displacement seismograms. The inversion assumes attenuation with a t* (travel time divided by average Q) of 1 s for P waves and 4 s for SH waves. The Green functions were computed for simple layered source and receiver structures connected by geometric spreading for a deeper ak135 Earth model[62]. The velocity structure at the source included a water layer overlying a half space with Vp = 6.0 km/s, Vs = 3.55 km/s and ρ = 2.67 g/cm³. The source was fixed at the epicentre derived from our study of the local seismicity described below. For the inversion, 16 P waves and 8 SH waves that provided good quality waveforms were chosen. The point source inversion supports a shallow strike-slip earthquake with a centroid at 10 km depth, supporting a mechanism of slip/dip/rake of 124/84/175 or 214/85/5 for the nodal plane. The conjugate fault matches the AIFS nicely, which was deduced from mapping and seismic reflection data. Later, the waveform data were used to derive the coseismic slip distribution. Today, the most robust finite-fault models are estimated by simultaneous inversion of seismic and geodetic data (GPS and/or InSAR); however, for earthquakes occurring at offshore settings there are neither geodetic nor InSAR data available. For these settings, like the case of the January 2016 Alboran Sea earthquake, the finite-fault slip distribution can only be determined through seismic data. Due to good data availability, seismic waveforms are still the most important data for studying coseismic slip. We used the iterative deconvolution method of Kikuchi and Kanamori[63] in the 0.01 to 1 Hz frequency band to

determine the slip distribution of the mainshock. In this method, the fault model is parameterised to have a rupture front that spreads over a grid of point-sources discretized in 12 × 8 subfaults of 5 km by 3.5 km. The moment rate function for each subfault was expressed by 5 triangle functions of 1.5 s duration and overlapping in time. The model assumes that the rupture consists of a propagating rupture front with slip accumulated in the wake of the rupture front passage. For the rupture speed Vr we tested a range of values between 1.5 and 4.5 km/s and finally found that a slow rupture velocity of ~2 km/s showed a minimum variance. We tested a number of different source parameterisations (including size of the rupture area) and found that the slip pattern remained robust. This suggests a shallow slip (<15–20 km) with a slip maximum near the epicentral area and the largest slip of ~100 cm at 10 km depth, where the point source inversion located the centroid. Our inversion shows similar features to the ones obtained by Buforn et al.[13]. To the south, the inversion suggested that the slip propagated very close to the seabed, perhaps nurturing surface rupture (Supplementary Fig. 3). Further, the slip inversion indicated that slip occurred towards the north. This feature is supported by regional waveform data from Spain, showing that the azimuthal distribution of apparent source times favours the shortest rupture for stations to the north of the rupture zone, and hence corroborates a northward rupture propagation as indicated by the slip inversion (Supplementary Fig. 4).

**Regional centroid moment tensors.** The determination of focal mechanisms by inversion of waveforms followed the grid-search procedure of Herrmann et al.[64]. Waveforms were obtained from the Instituto Geográfico Nacional (IGN)[15] and converted to velocity and rotated to radial, transverse and vertical components. Next, the data were bandpass filtered between 0.02 and 0.06 Hz to evaluate their quality. We selected waveforms that showed retrograde motion for the fundamental model Rayleigh wave, good signal to noise ratio, and finite signal duration.

The grid search technique takes samples over strike, dip and rake angles in 5 degree increments, and source depth in 1 km increments, in order to determine the shear-dislocation (double couple) that best fits the observed data. A feature of the implementation of the grid search is an efficient method for adjusting the predicted waveforms for time shifts that arise due to uncertainties in the assumed origin time and epicentral coordinates, the sampling of Green's functions with distance, and differences between the actual wave propagation and that of the 1-D model used. We tried several existing models, and the one that produced the best data fits was the WUS (Western US) model[64] (Fig. 5, and Supplementary Table. 1). In the Supplementary Fig. 8 we include, for every moment-tensor solution, detailed information with the number of stations used, processing parameters, modelling results and data fit.

**Re-location procedure of the seismic sequence.** Catalogue data available on the Instituto Geográfico Nacional (IGN, Spain) website (https://www.ign.es/ign/layout/sismo.do) were used to determine epicentral locations and focal depth for over 225 local earthquakes with $M_L > 3$. The non-linear oct-tree search algorithm Non-LinLoc[65] was used to calculate the focal parameters[61]. Travel times in the model were calculated using the finite-difference solution to the eikonal equation with a grid spacing of 2 km. The oct-tree algorithm provides more reliable information on location uncertainties than linearised inversions by exploring the probability density functions (PDF) of each individual event. The maximum likelihood location is chosen as the preferred location. For each event, Non-LinLoc estimates a 3D error ellipsoid (68% confidence) from the PDF scatter samples. Station statics account for localised deviations from the a priori model and are determined from the average residual at a station. For the inversion, the focal depth search was limited to depth >2 km, and thus rare cases of water quakes were avoided. From a previous study using an amphibious network we know that seismicity occurred at crustal levels[37]. We therefore restricted the focal depth of the crustal levels (<33 km). Travel times were calculated using the 1D local lithospheric velocity model derived from the amphibious network of Grevemeyer et al. (2015)[37] (i.e. ocean bottom seismometers and land stations), covering the Alboran Basin, Rif and Betics. Further, we corrected the 1D local lithospheric velocity model for effects caused by 3D propagation in a heterogeneous setting by introducing station correction terms. We only included stations used in Grevemeyer et al.[37] in 2010. Fortunately, the geometry of the permanent network has changed very little over the last 5 years. Therefore, we could use 60–80% of the seismic stations reported in the IGN catalogue (Fig. 5a). The station terms compensate for differences in the velocity structure caused by structural heterogeneity between the onshore and offshore domain, and hence provide an approximation of the 3D velocity structure. The main impact of station corrections is the eastward shift of the original locations reported by the IGN. Thus, relocated earthquakes occur 10–15 km eastward of the IGN located seismicity, providing an excellent spatial correlation between the AIFS imaged by bathymetric and seismic reflection data and the seismicity.

**Coulomb failure stress transfer modelling.** The Coulomb failure stress change was calculated for dislocations in an elastic half-space[66] and on slip planes (receiver faults henceforth) with a given strike, dip and rake[41,67]. The Coulomb failure stress change ($\Delta CFS$) is given by: $\Delta CFS = \Delta \tau_c - \mu' \Delta \sigma_n$, where $\Delta \tau_c$ is the change in shear stress (positive in the direction of the fault slip), $\Delta \sigma_n$ is the change in normal stress (positive in unclamping of the fault), and $\mu'$ is the apparent friction coefficient of

the fault. A positive increase in the Coulomb failure stress transfer in an area is interpreted as meaning that a fault plane located in this area has been brought close to failure, whereas if it is negative the interpretation is the opposite (i.e., relaxed). In the models, we have assumed a $\mu'$ of 0.4, a typical Poisson ratio of 0.25 and a Young modulus of $8 \times 10^5$ bar (last two parameters compute for a shear modulus of $3.2 \times 10^5$ bar). Although values of $\mu'$ lower than 0.4 might be appropriate on strike-slip faults[68,69], its variation only modestly modifies the stress distribution around a fault[41,70]. The modelling was carried out using the Coulomb 3.4 software.

The source fault is the fault plane that is displaced during the earthquake. In the $\Delta CFS$ modelling, we considered the source fault to be the one that mimics the coseismic slip determined from inversion of the teleseismic waveforms, and the rupture plane corresponding to a section of the AIFS. This section is bent and extends, from south to north between −3.61°W/35.75°N and −3.78°W/35.19°N, and bends at −3.72°W/35.59°N (location of the epicentre). North of the releasing bend, the source fault extends for 20 km along the Central segment and strikes 031° N. South of it, the South segment extends for 45 km and strikes 007°N. Both sections are vertical and have a rake of 5° (left-lateral strike-slip with a reverse component). To mimic the slip model presented in this work (Supplementary Fig. 3), the source fault was divided into 1408 sub-sources, ~1-km-wide and 1.5-km-long, each with its estimated slip. The automatic seismic moment and moment magnitude calculated by the Coulomb software gives a seismic moment ($M_0$) of 8.79 $10^{18}$ Nm and a moment magnitude ($M_w$) of 6.60. These results are slightly larger than those obtained from the seismological data.

The $\Delta CFS$ was calculated on two different types of receiver faults. The strike, dip and rake of these faults were established based on information provided by the focal mechanisms of aftershocks recorded in the area. The first type are left-lateral strike-slip faults striking 210°N, dipping 90° and with a rake of 0°, which correspond approximately to the Central Al-Idrissi Fault segment that trends parallel to the Alboran Ridge Fault System and to the focal mechanisms solutions for some aftershocks. The second type of receiver faults are reverse faults striking 070°N, dipping 45° and with a rake of 85°, which coincide with the direction of the South Al-Idrissi Fault segment and the moment-tensor solutions of some aftershocks.

**Reporting summary.** Further information on research design is available in the Nature Research Reporting Summary linked to this article.

## Data availability

The source data underlying Figs. 1, 2, 5, and 7 and Supplementary Fig. 7 are provided as a Source Data file. Data associated with this paper, such as topography, bathymetry and seismological data are available. TOPOGRAPHY: NASA JPL (2013). NASA Shuttle Radar Topography Mission Global 1 arc second [Data set]. NASA EOSDIS Land Processes DAAC. https://doi.org/10.5067/MEaSUREs/SRTM/SRTMGL1.003. BATHYMETRY: EMODnet Bathymetry Consortium (2018): EMODnet Digital Bathymetry (DTM). https://doi.org/10.12770/18ff0d48-b203-4a65-94a9-5fd8b0ec35f6. REGIONAL CENTROID MOMENT TENSORS: The repository of regional centroid moment tensors can be downloaded at this link: https://digital.csic.es/handle/10261/177887, https://doi.org/10.20350/digitalCSIC/8623. DATA FROM IRIS DATA CENTRE: This work included data from the II, IU, GE, GT, and G seismic networks obtained from the IRIS data centre. https://doi.org/10.7914/SN/II; https://doi.org/10.7914/SN/IU; https://doi.org/10.14470/TR560404; https://doi.org/10.7914/SN/GT; https://doi.org/10.18715/GEOSCOPE.G. DATA FROM THE IGN CATALOGUE: http://www.ign.es/web/en/ign/portal/sis-catalogo-terremotos.

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

## Acknowledgements

This research was supported by projects CGL2011-30005-C02-02, CTM2015-70155-R and Acción IDRISSI (MINECO/FEDER), shiptime on board the R/V Sarmiento de Gamboa and R/V Angeles Alvariño, EuroFleets-2 grant agreement 312762, and access to IFREMER AUVs AsterX and IdefX (France) through Ocean Facilities Exchange Group (OFEG). We also thank the Instituto Geografico Nacional (IGN, Spain) for providing seismological data. H.P. acknowledges the EU Horizon 2020 programme under grant agreement No H2020-MSCA-IF-2014 657769. E.d'A. acknowledges the French programme Actions Marges, the LabexMER (ANR-10-LABX-19-01) and the EuroFleets grant agreement 228344.

## Author contributions

E.G. conceived the study, led the EVENT, TOPOMED-GASSIS and SHAKE marine experiments, designed the research and wrote the paper with contributions and edits from all authors. I.G. relocated the seismicity, calculated the mainshock focal mechanism and participated in writing the paper. H.P. modelled the Coulomb failure stress and together with R.B., S.M.L., L.G.P., C.L.I., S.D., A.C., S.C. and M.C. participated in the data acquisition at sea and completed the data processing and interpretation. A.V. calculated the moment-tensor solutions of the aftershocks, and Y.K. contributed in data analysis and interpretation. E.d'A. and A.R. acquired and provided new bathymetric data from the Moroccan Margin. C.R. led the TOPOMED project, contributed to interpretation and participated in writing the paper.

## Additional information

**Competing interests:** The authors declare no competing interests.

