## [Peer Review File · Nature Communications]

Reviewers' comments:

Reviewer #1 (Remarks to the Author):

General comments

The paper's main claim is that a young fault system in the Alboran Sea, in between the Iberian Peninsula and North Africa, may pose sometime in the future a major catastrophic hazard due to the associated seismicity.

The paper is based on a robust dataset and duly takes into account previous work of relevance in the area.

The paper is well constructed and may attract the interest of others in the expert community and beyond. However, some interpretations look somehow forced for them to fit with the authors' views. Alternative explanations, leading to less worrying scenarios, are in a way neglected or minimised. I suggest lowering the catastrophic tone of the last paragraphs and remove, or at least modulate or identify, too speculative components.

In other words, the main strength of the paper is the rather lengthy (and sometimes wordy) descriptive part and the way authors connect different evidences and inferences on the expression, significance and dynamics of AIFS.

Consistency amongst figure labels, acronyms and the main text needs to be worked out.

High-resolution copies of the non-interpreted seismic reflection profiles showed in the paper must be fully available online (i.e. as supplementary material), and this must be mentioned in the manuscript.

Please, see specific comments below for further details.

Specific comments

It may look strange to some readers that the authors refer to "a large continental fault" within a marine basin, the Alboran Sea. I suggest they clarify this point (i.e. by providing an explanation for non experts)

Lack of adjustment between the way some elements are referred in the main text and in the figures, which rends reading and understanding more difficult than it should.

Acronyms in figures and text: sometimes they appear without dots (e.g. AIFS, TASZ, ARF, NSF, EBSZ,.. in Fig. 1) while in other occasions they appear with dots (e.g. E.A.B., FZ vs. F. in Fig. 2), or in other different forms in the same figure (e.g. AIFS and Al-Idrissi F.S.in Fig. 2). I suggest using a uniform criterion for acronyms. Also check consistency of capitalization in figures.

Too many acronyms, not always used consistently, make reading text and figures rather hard.

In some passages of the paper it looks as if there is some degree of over interpretation for the observations (and the lack of them) to fit with the authors' ideas, thus ruling out alternative explanations. See detailed comments further down.

L40: (inset in Fig. 1).

L42: and reverse fault systems (Fig. 1) - Thrust faults in Figure 1

L43: NW–SE trending convergence or NW–SE to/or NNW-SSE trending convergence?

L44-L45: What does it mean “Shallow to deep” earthquakes? All earthquakes are “shallow to deep”. Is this actually necessary?...of small to moderate magnitude characterize “the local seismicity”. I would be better: “Seismicity in the study area is characterized by earthquakes of small to moderate magnitude”.

L45-L49: Rewrite, better English is needed. First subject and verb, then examples.

L50: Besides deep (> 100 km) seismicity in the West Alboran Basin.... Very few if any earthquake seem to be “deep” in the West Alboran Basin s.str. according to Fig. 1.

L50-L51: Add RGB label to Fig. 1.

L51: an ~80 km wide NE-SW seismic zone... or NNW-SSE?

L54-L55: ...connecting the Eastern Betic Shear Zone (EBSZ) in Spain to the Rif in Morocco.

In Figure 1 the labels are Iberian Peninsula and North Africa, while in the main text you refer to Spain and Morocco, which are political entities. Why is this? I would suggest you take the option that is more relevant according to the paper contents.

L56-L58: Make this sentence shorter, stay to the essentials.

L60-L61: Names and acronyms do not fit with those in Fig. 1 and they have to.

L68-L69: I guess your present something else than a map (and associated crustal structure). Please rewrite.

L69: Is such approach actually “unique”?

Make labels in all figures fully consistent. At present they are not (e.g. Carboneras F. in Fig. 2, and CF in Fig. 1, or Morocco vs. North Africa in Fig. 2. One thing, one name, not two or three different ones).

L81: NE-SW... or NNE-SSW?

L85-L87: Near where the destructive 1910 Adra earthquake (estimated Mw ~6.1), the historical 1804 (MSK Intensity VIII) and the recent 1993-1994 seismic crises occurred (Fig. 2a)

You should mention “Adra” the first time you refer to the 1910 earthquake (L46). Some information is reiterative (L46). What are these “recent 1993-1994 seismic crises”? I have not read anything about them previously in the manuscript. “Recent” is not needed as you already wrote “1993-1994”. Major earthquake codes (AD10, AD93, AL94...) in Fig. 2 do not fit with the way they are referred to in the main text. This makes reading cumbersome. Needs rewriting.

Also, information on these events should be packed in one single place within the manuscript, instead of scattered in different places.

L100: “circulation” of what? Through what? Along what? Bottom circulation of across seafloor fluid flow? Provide dimensions of pockmarks. How can you demonstrate that bottom current scouring around hard elements or objects instead of other hypotheses is not the cause of these pockmarks?

L106-L107: How do you know that the “area referred to as the Djibouti Plateau FZ” is of volcanic nature? Dredging of the flank of La Herradura seamount of the Djibouti group has provided granitoids (unpublished). A reference is needed to sustain your sentence, or you modify it.

The limits between north, central and south AIFS should be clearly marked in the relevant figures.

L119: Remove “yet”.

L132: Say what makes the difference amongst these two crustal domains.

L141: allows.

L202-L206: The lack of aftershocks in the southernmost central AIFS following the increase of stress is certainly intriguing. The explanation provided seems an easy outlet. Could the authors provide other alternative explanations?

L213-L218: This is to some extent speculative. Could the authors better sustain or develop their interpretation?

L232: I guess “the northern Spanish Margin” is in the Cantabrian Sea. Why you don’t say simple “the northern Alboran Sea margin”?

L233-L237: Could you rule out the possibility that events simply remained unreported?

L237-L248: The comparison with San Andreas, the Dead Sea and the North Anatolian fault systems is probably going too far. Geodynamic settings, slip rates, dimensions and levels of seismic activity are markedly different. At least some words of caution would be required if the authors want to keep such, in my opinion, risky comparisons. The authors state that the AIFS “could eventually develop into a large-scale continental plate-boundary fault zone”, but it could be equally possible that such evolution does not take place, and it continues as a “baby” fault system, like others. The drivers of accommodation and eventual propagation take place at such a large scale that they cannot be properly interpreted from a relatively minor feature such as AIFS. I suggest removing highly speculative elements from the text or, alternatively, stress that they are speculative.

L253-L258: Convert into two or more sentences. Again, be careful with speculation (e.g. “we would end up with a rupture of 160 km”). This is nothing good for the paper.

L260-L263: Needs rewriting in better English.

L263-L268: Not sure I fully understand this sentence. Is the “cascade-like activation” something you foresee for the future, or is it how you see the set of known past events? Also, consider the likelihood of events that could have not been detected.

L269-L274: I suggest you lower the tone. There are already too many catastrophe sellers.

Reviewer #2 (Remarks to the Author):

Remarks to the author:

This is a very interesting, well written, and well documented research paper. It combines existing knowledge on tectonic evolution with analysis by the authors of various new data sets and presents a convincing case of active left-lateral fault systems, occurring from south to north across the Alboran Sea, that line up to form the Al-Idrissi Fault System (AIFS). With various seismic

sections and the aftershocks of the large 2016 earthquake the authors demonstrate that the AIFS cuts through the continental crust of the Alboran basin. The lithosphere of the strongly stretched Alboran basin is rather thin and, being above the subduction wedge for 10s of Myr, it is also expectedly warm. Therefore, the authors could have safely speculated that the AIFS likely starts cutting through the entire Alboran lithosphere. The authors present a convincing case of a nascent continental fault system that defines the plate boundary between Europe (to the east and north) and Nubia (to the west and south). For long, the location of this plate boundary has been enigmatic and subject of debate but with their findings the authors provide the compelling evidence that the AIFS acts as the plate boundary across the Alboran. This finding is a crucial step forward for unraveling this complex tectonically complex region and is provides a clear direction for further investigations of the cause of earthquakes in the region for which the authors present the interesting case that the magnitude of earthquakes along the AIFS may increase to larger than M7 once the AIFS is entirely broken.

The focus of the paper is much on the crustal structure and earthquake generating potential associated with the AIFS. The identification of the regional plate boundary fault system is equally exciting news. The paper substantiates a clear direction for the important issue of where the plate boundary is in the region which not only gives guidance to research into the seismo-tectonic but also to the past tectonic evolution of the region. This latter aspect is, however, a bit undertreated by the authors and I would like to suggest to briefly review (in the Introduction) the various literature propositions made for the tectonic role of the TASZ. It has only been propounded as plate boundary by some and not by most researchers, e.g. see Vernant 2010 for several literature propositions. The Bird-proposition of a plate boundary running from the Atlantic through northern Morocco seems to be mostly used regionally (e.g. Neres et al. 2016) as well as in the latest global plate kinematic models for the last few million years (Argus et al. 2011) as successor of the NUVEL model. I suggest to enhance this discovered role of the AIFS which is now only touched upon in the last few sentences of the paper (271-274).

I have only few minor comments and advise publication after minor revision.

Wim Spakman

Title: replace 'embryonic' by 'incipient' (it is beyond embryonic')

Continental -> 'plate boundary' ; this is more to the point while 'continental' is rather general. Although the Alboran comprises strongly stretched continental crust, it can hardly be taken as typical for 'continental'

37: continental -> plate-boundary

42-43: Crustal deformation cannot be 'mainly' explained by relative plate convergence (ref 14).

Suggest to replace by:

... fault systems within the overall plate tectonic setting of NW-SE trending convergence (Fig. 1).

51 ref 13 (Wortel and Spakman 2000) is not appropriate because they did not state that. Better replace by Gutscher 2014 and/or Spakman and Wortel 2004.

58: "plate subduction activated" is vague.

Suggestion: " ... by NNE directed dragging of the RGB slab by the Nubia plate in the past ...'

This also improves the flow to the next sentence.

64-68: This interrupts a bit the flow of the introduction. Suggest to move to line 49, or to provide a bridge with the previous paragraph.

68 'map' -> 'structural map'

77 'might' -> 'may'

81 "NE-SW" -> "NNE-SSW" ; the azimuths you report later belong to NNE-SSW

247 Boundaries between plate are defined by lithosphere cutting faults, not by crustal boundaries. You could make the case here that the AIFS may as well cut the thin hot Alboran lithosphere (likely by fault creep).

Reviewer #3 (Remarks to the Author):

Review Gracia et al., Nature Communications

As the authors say, the occurrence of this intense earthquake sequence on a young continental fault is a great opportunity to understand the evolution of strike-slip fault systems. Besides the treatment of a hot topic, the study is based on excellent data from different bathymetric and seismic campaigns, as well as earthquake recordings. The conclusions are well-thought and consistent, interesting for a broad readership, and clearly worth a place in a top scientific journal. I recommend publication in Nature Communications after minor revision.

In the following I will list a number of criticisms and comments to the authors. Some of them should be definitely addressed, I believe, while others may be more optional and arise from my subjective point of view. Some aspects of this study appear less solid than others, and I think the authors might try to clarify or omit these points, to avoid that their general conclusions may become vulnerable, too.

1) The article is constructed around the characterization of the Al Idrisi Fault (AIF) as an embryonic, initial-stage, young, or nascent fault system. I think this is a very appealing and useful model that may explain many features of the region, and it sufficiently justifies the special relevance of analysing this fault. However, some numbers, for example the accumulated slip, may be relevant to explain this concept. For example, the Alboran Ridge (as a bathymetric feature) appears offset by about 10 km by the fault (Fig. 2), which would be consistent with the age (Late Pliocene to Quaternary, e.g. 3 Ma) and plausible slip rates (e.g. 3 mm/a). This, by the way, is a rather normal ratio between fault length and offset. So it seems ok to call the fault initial-stage or young, but embryonic or nascent may be exaggerated.

2) You predict that the AIF will develop into a large-scale continental plate-boundary; I think also this point deserves more explanation: The AIF is trending close to N-S, approximately perpendicular to the overall direction of the Nubia-Iberia plate boundary. So please describe the trace of the plate boundary that would involve the AIF, and the way that tectonic deformation at this boundary would work.

3) Previous work on the AIF in seismic and bathymetric studies is acknowledged properly and well embedded in the manuscript, but previous seismological studies are not described with sufficient detail. Buforn et al., 2017 and Kariche et al., 2018 are merely mentioned once in the introduction. These studies have significant overlap with your work, and you should compare data, methods and results.

4) Not enough information is provided on regional moment tensor inversion regarding stations (number of stations, distances, azimuthal gap), earth model (no information at all) and results (please show some waveform fits for some examples of the smaller earthquakes as supplementary material).

5) The result from finite fault modelling is a strongly asymmetric rupture that propagates about 20 km southwards, at low rupture speed of 2 km/s. This is the opposite from the result proposed in Buforn et al., an asymmetric rupture that propagates about 20 km northwards, at high rupture speed of 3 km/s. This dissonance is noteworthy because both studies use the same method and inversion code (Kikuchi and Kanamori), same type of data (teleaseismic P and SH waves from IRIS) and also similar models (both incorporating the water layer, which is good). Since Buforn et al. is the first study, it corresponds to this study to comment on these differences. Unfortunately, Buforn et al. do not provide waveform fits. On the other hand, waveform fits in this study (supplementary Figure 2) do not support the proposed rupture model convincingly. In particular, the longest observed SH pulses (MBO) and P pulses (DBIC, I'd say) are located towards south, which seems to be at odds with the proposed southward propagating rupture. The northwards propagation by Buforn et al. is supported by the directivity function of long-distance Rayleigh waves. In this study,

a weird attempt to characterize directivity from regional data has been done (supplementary figure 4). The proposed azimuth to make the cosine function fit the data is 0 degrees, i.e. north. This actually fits the shorter arrivals towards north, contradicting the southwards rupture propagation just inferred one paragraph before from teleseismic modelling. My personal balance of evidence is that rupture likely propagated to the north. My suggestion would be to omit this point, because it is really weak, and actually not necessary for the main conclusions of this study. If you want to keep rupture directivity, then please also report what exactly is shown in supplementary figure 4 (Vertical P-waves?), and report the parameter you use to fit the equation given in the figure caption.

6) I'm really unable to identify the Moho in supplementary figure 1, and in consequence the Moho step across the AIF, so I cannot reach to the conclusion that two different crustal domains are observed across the AIF. In particular, energy vanishes around 10.5 s TWT along the entire profile, not only the SE part. I suppose that this is due to the representation, and you can see the Moho in the original, full-resolution seismic section. Is there any other study that supports the Moho step? Receiver functions from nearby coastal stations (Mancilla and Diaz, Tectonophysics, 2015) rather suggest it's the other way round: The crust is much thinner to the east of the intersection between the AIF and the Moroccan margin, than to the west. Please explain.

Two minor comments:

Line 82: The AIF "accommodates a total rate of ~3.8 mm/yr of relative motion". This appears speculative, since this number is taken from a GPS study, but the AIF is an offshore fault along its entire length. Please phrase it more carefully.

Line 332: slip/dip/rake of 124/84/175 and 214/85/5 are not conjugate faults (at ~60 degrees, for typical friction parameters in geology), but the orthogonal fault and auxiliary planes of the symmetric point source mechanism (at 90 degrees).

Reviewers' comments:

Reviewer #1 (Remarks to the Author):

General comments

The paper's main claim is that a young fault system in the Alboran Sea, in between the Iberian Peninsula and North Africa, may pose sometime in the future a major catastrophic hazard due to the associated seismicity.

The paper is based on a robust dataset and duly takes into account previous work of relevance in the area.

The paper is well constructed and may attract the interest of others in the expert community and beyond. However, some interpretations look somehow forced for them to fit with the authors' views. Alternative explanations, leading to less worrying scenarios, are in a way neglected or minimized. I suggest lowering the catastrophic tone of the last paragraphs and remove, or at least modulate or identify, too speculative components.

In other words, the main strength of the paper is the rather lengthy (and sometimes wordy) descriptive part and the way authors connect different evidences and inferences on the expression, significance and dynamics of AIFS.

Consistency amongst figure labels, acronyms and the main text needs to be worked out. High-resolution copies of the non-interpreted seismic reflection profiles showed in the paper must be fully available online (i.e. as supplementary material), and this must be mentioned in the manuscript.

Please, see specific comments below for further details.

Thank you for your general comments, which are much appreciated. We will use them to improve the content of our ms. We agree in lowering the tone in the references to certain interpretations and hazard estimations.

We will also make the figure labels, acronyms and main text consistent. Non-interpreted seismic reflection profiles have been included in Supplementary Figure 1 and will be mentioned in the ms.

Specific comments

1) It may look strange to some readers that the authors refer to "a large continental fault" within a marine basin, the Alboran Sea. I suggest they clarify this point (i.e. by providing an explanation for non-experts)

The Alboran Sea is a Neogene continental basin located in the westernmost Mediterranean and generated by a subduction process, which has been progressively retreating from the Balearic Islands to the Straits of Gibraltar (Spakman & Wortel, 2004; Spakman et al., 2018). As a result of this process, different domains have been identified that correspond to the different elements composing a subduction system: the accretionary wedge (located in the Gulf of Cadiz), the fore-arc basin (located in the West Alboran), the volcanic arc (East Alboran) and the back-arc (Algero-Balearic Basin).

Below, we show a figure that summarizes the different crustal domains of the Alboran Sea.

Rev. Fig. 1: Bathymetric map including the main faults, showing the different domains associated with a Gibraltar subduction system. Black lines: faults; Red dashed lines: volcanic areas. Modified from Gómez de la Peña et al., 2018 (Tectonics)

2) Lack of adjustment between the way some elements are referred in the main text and in the figures, which renders reading and understanding more difficult than it should. Acronyms in figures and text: sometimes they appear without dots (e.g. AIFS, TASZ, ARF, NSF, EBSZ,... in Fig. 1) while in other occasions they appear with dots (e.g. E.A.B., FZ vs. F. in Fig. 2), or in other different forms in the same figure (e.g. AIFS and Al-Idrissi F.S. in Fig. 2). I suggest using a uniform criterion for acronyms. Also check consistency of capitalization in figures.

Too many acronyms, not always used consistently, make reading text and figures rather hard.

We thank the reviewer for identifying the inconsistencies in the acronyms. We realize that there are numerous abbreviations and I apologize that some of them appear different either in the text and/or in the figures. Now, we have unified the acronyms, and we are using initial letters without points. For example, we have modified Al-Idrissi F.S. to AIFS.

3) In some passages of the paper it looks as if there is some degree of over interpretation for the observations (and the lack of them) to fit with the authors' ideas, thus ruling out alternative explanations. See detailed comments further down.

We thank you for your comments. We agree that in some cases, especially in the Discussion, there might be some over interpretation. We apologize for this, and we now propose more realistic scenarios.

L40: (inset in Fig. 1).

OK, this has been modified as requested.

L42: and reverse fault systems (Fig. 1) - Thrust faults in Figure 1

We thank the reviewer. We have now changed it to "thrust faults".

L43: NW-SE trending convergence or NW-SE to/or NNW-SSE trending convergence?

We agree with the reviewer. We have now changed it to NW-SE to NNW-SSE trending convergence. (now in lines 43-44)

L44-L45: What does it mean "Shallow to deep" earthquakes? All earthquakes are "shallow to deep". Is this actually necessary?...of small to moderate magnitude characterize "the local seismicity". It would be better: "Seismicity in the study area is

characterized by earthquakes of small to moderate magnitude”.

Thanks for your comment. Yes, we agree that the “Shallow to deep” sentence is not necessary. We have changed the sentence as you suggested: “*Seismicity in the study area is characterized by earthquakes of small to moderate magnitude*”. (now in line 45)

L45-L49: Rewrite, better English is needed. First subject and verb, then examples.

We have rewritten the sentence following your comments: “*Large historical and instrumental earthquakes have occurred in the region, such as the 1804 and 1910 (MSK Intensity VIII to X), and the M_w 6.0, 26 May 1994 and the M_w 6.3, 24 February 2004 Al-Hoceima earthquakes*”. (now in lines 46-48)

L50: Besides deep (> 100 km) seismicity in the West Alboran Basin.... Very few if any earthquake seems to be “deep” in the West Alboran Basin s.str. according to Fig. 1.

We agree and we thank you for your comment. The seismic activity in the Alboran Sea ranges from shallow to intermediate seismicity. Thus, the seismic activity recorded in the West Alboran Basin can be considered as intermediate, not deep. (now in line 55)

L50-L51: Add RGB label to Fig. 1.

As the RGB label is only mentioned once in the entire ms, we decided to remove the acronym RGB, and leave the complete text: “*Rif-Gibraltar-Betics*”. (now in line 56)

L51: an ~80 km wide NE-SW seismic zone... or NNW-SSE?

The Trans-Alboran Shear Zone (TASZ) comprises a wide fault zone that mainly trends NE-SW. However, if we focus only on the Al-Idrissi Fault System (AIFS), then the main trend is NNE-SSW. In the ms, we have kept “*NE-SW trending seismic zone*”. (now in line 56)

L54-L55: ...connecting the Eastern Betic Shear Zone (EBSZ) in Spain to the Rif in Morocco. In Figure 1 the labels are Iberian Peninsula and North Africa, while in the main text you refer to Spain and Morocco, which are political entities. Why is this? I would suggest you take the option that is more relevant according to the paper contents. We thank the reviewer for his/her wise comment. We apologize for our mistake referring to specific political entities, instead of using geographic/geological terminology. Now, we have changed the text accordingly and the modified sentence is: “*... connecting the Rif (North Africa) to the Eastern Betic Shear Zone (SE Iberian Peninsula)*”. (now in lines 59-60)

L56-L58: Make this sentence shorter, stay to the essentials.

We rewrote the sentence as follows: “*A recent work combining geological, geodetic and 3D numerical modelling¹⁷ demonstrates that crustal deformation in the Alboran Sea, induced by NNE-directed dragging of the RGB slab by the Nubia plate in the past 8 Myr, is still active*”. (now in lines 63-66)

L60-L61: Names and acronyms do not fit with those in Fig. 1 and they have to.

Thanks for the comment. We kept AIFS as the acronym, but the rest of the faults that are not the focus of the paper are written by their full names. Now, the new sentence is as follows: “*Such recent deformation has been documented, for example, along the Yusuf Fault, Carboneras Fault System and especially along the AIFS and associated structures of the Rif and the Eastern Betics Shear Zone (Figs. 1 and 2b), which comprise the main fault systems of the TASZ (Figs. 1 and 2)*”. (now in lines 67-70)

L68-L69: I guess you present something else than a map (and associated crustal structure). Please rewrite.

Thanks for your comment. The sentence has been completely rewritten, and the result is as follows: “*Here we present a new and comprehensive geological and geophysical dataset of the entire AIFS*”. (now in line 71)

L69: Is such approach actually “unique”?

We fully agree with the Reviewer, as the approach is not actually “*unique*”. We have removed this word from the ms. The new sentence is: “*We adopted a multi-scale approach, including shipboard multibeam bathymetry...*” (now in line 72)

Make labels in all figures fully consistent. At present they are not (e.g. Carboneras F. in Fig. 2, and CF in Fig. 1, or Morocco vs. North Africa in Fig. 2. One thing, one name, not two or three different ones).

Figure 1 is very dense and there is not enough space to include the full names of all the faults. This is the main reason why in Figure 1 we include only the acronyms of the faults. *In the caption of Figure 1*, all acronyms are defined by their full name. In the text of the ms we refer to the faults by their full names: Carboneras Fault System and Yusuf Fault System, with the exception of the AIFS (Al-Idrissi Fault System), in which we kept its acronym.

L81: NE-SW... or NNE-SSW?

Thank you. The correct trend is NNE-SSW. (now in line 83)

L85-L87: Near where the destructive 1910 Adra earthquake (estimated Mw ~6.1), the historical 1804 (MSK Intensity VIII) and the recent 1993-1994 seismic crises occurred (Fig. 2a). You should mention “Adra” the first time you refer to the 1910 earthquake (L46). Some information is reiterative (L46).

We have modified the text and included “1910 Adra earthquake”. (now in lines 47 and 88)

What are these “recent 1993-1994 seismic crises”? I have not read anything about them previously in the manuscript. “Recent” is not needed as you already wrote “1993-1994”. The 1993-1994 seismic crises have been documented in the following article:

- Stich, D., Alguacil, G. and Morales, J. (2001). “The relative locations of multiplets in the vicinity of the Western Almería (southern Spain) earthquake series of 1993-1994”, *Geophysical Journal International*, 146, 801-812.

We have removed “*recent*” from the sentence in L86. (now in line 89)

Major earthquake codes (AD10, AD93, AL94...) in Fig. 2 do not fit with the way they are referred to in the main text. This makes reading cumbersome. Needs rewriting. Also, information on these events should be packed in one single place within the manuscript, instead of scattered in different places.

The major earthquake codes AD10, AD93, AL94, DJ04, etc. are all included in the caption of Figure 2. They are also included in the main text of the manuscript: DJ04: Djibouti plateau 1804; AD10: Adra 1910; AD93-94: Adra series 1993-1994; AL94: Al-Hoceima 1994; AL04: Al-Hoceima 2004. (now in lines 88-90)

L100: “circulation” of what? Through what? Along what? Bottom circulation of across seafloor fluid flow? Provide dimensions of pockmarks. How can you demonstrate that bottom current scouring around hard elements or objects instead of other hypotheses is not the cause of these pockmarks?

We were referring to “fluid flow circulation” through the seafloor. We have modified the text accordingly. Pockmarks are circular craters found in the seabed and caused by fluids (gas and liquids) erupting and streaming through the sediments. They are uncommon on the land surface, but they are very common on the ocean floor. Actually, pockmarks have been found worldwide in the oceans. In the 1960s, pockmarks were discovered offshore of the coasts of Nova Scotia (Canada) using a side-scan sonar. The craters were up to 150 m in diameter. Later, with advances in technology, high-resolution multibeam acoustic systems have been used to map pockmarks. Pockmarks are interpreted as the morphological expression of fluid flow leakage from active hydrocarbon systems or deep over-pressured petroleum reservoirs. The pockmarks identified near the AIFS are between 15 and 60 m in diameter. Bottom current scouring was definitively not the cause of these pockmarks. Below are some references to pockmarks, the second one (in bold) is referred to in the ms:

- Pilcher, R., Argent, J. (2007). Mega-pockmarks and linear pockmark trains on the West African continental margin. *Mar. Geol.* 244, 15-32.
- **Andresen, K.J. & Huuse, M. (2011). ‘Bulls-eye’ pockmarks and polygonal faulting in the Lower Congo Basin: Relative timing and implications for fluid expulsion during shallow burial. *Mar. Geol.* 279 (1-4), 111-127. (now in lines 102-103)**
- Riboulot, V, et al. (2013). Sea-level change and free gas occurrence influencing a submarine landslide and pockmark formation and distribution in deepwater Nigeria. *EPSL* 375, 78-91.
- Mahiques et al., (2017). An extensive pockmark field on the upper Atlantic margin of Southeast Brazil: spatial analysis and its relationship with salt diapirism. *Heliyon* 3, 21 pp.

L106-L107: How do you know that the “area referred to as the Djibouti Plateau FZ” is of volcanic nature? Dredging of the flank of La Herradura seamount of the Djibouti group has provided granitoids (unpublished). A reference is needed to sustain your sentence, or you modify it.

The area referred to as the Djibouti Plateau FZ is composed by a magmatic arc crust and magmatic intrusions, on the basis of the following references: Duggen et al., *EPSL* (2004) and Gómez de la Peña et al., *Tectonics* (2018). These two references have now been added to the bibliography. We have modified the sentence to: “...the Djibouti Plateau FZ is composed by a magmatic arc crust and magmatic intrusions” (now in lines 109-110)

The limits between north, central and south AIFS should be clearly marked in the relevant figures.

The boundaries between the North (N), Central (C), and South (S) segments are located in Figures 2b, 5a, 6 and 7.

L119: Remove “yet”.

Thanks. We have removed “yet”, now the sentence is: “...not appear to be fully connected” (now in lines 122-123)

L132: Say what makes the difference amongst these two crustal domains.

Thanks for the reminder. Profile TM2 shows an evident difference in crustal thickness across the AIFS. The thickness of the West Alboran crust is ~8 s TWTT, while the thickness of the North African crust is ~10.5 s TWTT. This has been added to the manuscript and is also explained in the Supplementary Fig. 2. (now in lines 136-137)

L141: allows.

We have changed it to: “this makes it possible to...” (now in lines 145-146)

L202-L206: The lack of aftershocks in the southernmost central AIFS following the increase of stress is certainly intriguing. The explanation provided seems an easy outlet. Could the authors provide other alternative explanations?

We do not know if it is accumulating or it is released by creep or slow-slip. However, our data does not indicate any seismic release (i.e. we do not see any aftershocks in the southernmost central AIFS). Onshore, it may be easier to address this issue as we have the advantages of the GPS, which we do not have in marine areas. **(now in lines 225-228)**

L213-L218: This is to some extent speculative. Could the authors better sustain or develop their interpretation?

We disagree with the reviewer. The process that we mention refers to how fault systems grow: “*Short segmented faults generate small displacements and are thus associated with small magnitude earthquakes. With time, accumulation of seismic slip leads to a simpler fault geometry, and longer strike-slip fault zones with the potential to generate large magnitude earthquakes*”. Thus, we are not talking about interpretation or speculation, this description has been demonstrated by successive high-impact articles led by experts in active faulting and seismology. Below, we give some of the key references:

- Wesnousky, S. G. Predicting the endpoints of earthquake ruptures. *Nature* 444, 358-360 (2006).
- King, G. C. P. & Wesnousky, S.G. Scaling of fault parameters for continental strike-slip earthquakes. *Bull. Seismol. Soc. Am.* 97, 1833–1840 (2007).
- Biasi, G. P. & Wesnousky, S. G. Steps and Gaps in Ground Ruptures: Empirical Bounds on Rupture Propagation. *Bull. Seismol. Soc. Am.* 106(3), 1110–1124 (2016). **(now in lines 235-240)**

L232: I guess “the northern Spanish Margin” is in the Cantabrian Sea. Why you don’t say simple “the northern Alboran Sea margin”?

Thanks for your comment. We completely agree. We have modified the sentence to “Along the northern margin of the Alboran Sea”. **(now in line 254)**

L233-L237: Could you rule out the possibility that events simply remained unreported?

It is true that we cannot rule out the possibility that historical and instrumental earthquakes of low magnitude ($I_{\max} < 4$) that may have occurred in the Alboran Sea have gone unreported. In any case, they are not essential for the main aim of our work. The “*Instituto Geográfico Nacional*” (IGN, Seismology Section) has a complete catalogue of moderate to large historical earthquakes (i.e. years 1000 to 1899), which we used: http://www.ign.es/web/resources/sismologia/NERIES/query_eq/index.htm. In addition, for our work we also used the instrumental catalogue of the IGN web page: <http://www.ign.es/web/resources/sismologia/tproximos/prox.html#> **(now in lines 256-259)**

L237-L248: The comparison with San Andreas, the Dead Sea and the North Anatolian fault systems is probably going too far. Geodynamic settings, slip rates, dimensions and levels of seismic activity are markedly different. At least some words of caution would be required if the authors want to keep such, in my opinion, risky comparisons. The authors state that the AIFS “could eventually develop into a large-scale continental plate-boundary fault zone”, but it could be equally possible that such evolution does not take place, and it continues as a “baby” fault system, like others. The drivers of accommodation and eventual propagation take place at such a large scale that they cannot be properly interpreted from a relatively minor feature such as AIFS. I suggest

removing highly speculative elements from the text or, alternatively, stress that they are speculative.

Thanks for your comment, which we appreciate. In the text of the ms we have changed the original sentence “*the AIFS has the potential to generate larger events*” to “*the AIFS may have the potential to generate...*”. (now in lines 259-260)

We know the succession of large earthquakes from 1994, 2004 and the last one of 2016. So, the AIFS, despite being a young continental fault system, is growing and eventually “*could develop into a large-scale continental plate-boundary fault along the Trans-Alboran Shear Zone*”. (now in lines 264-266)

L253-L258: Convert into two or more sentences. Again, be careful with speculation (e.g. “we would end up with a rupture of 160 km”). This is nothing good for the paper. Thanks for your comment. We have modified the sentence and “softened” its content, for instance: “*eventually, it may result in a maximum rupture of 160 km*”. (now in line 280)

L260-L263: Needs rewriting in better English.

We appreciate your comment. We have rewritten the sentence, as suggested. The outcome is:

“*A sequence of historical (AD1804 and AD1910) and instrumental (1994, 2004 and 2016) earthquakes with estimated magnitudes ranging from M_w 5.9 to 6.4 has hit the Alboran Sea region in northern Morocco and southern Spain (Fig. 7)*”. (now in lines 283-285)

L263-L268: Not sure I fully understand this sentence. Is the “cascade-like activation” something you foresee for the future, or is it how you see the set of known past events? Also, consider the likelihood of events that could have not been detected.

Indeed, the “cascade-like activation” of fault segments towards the north and south refers to the set of known past events (from years 1804 to 2016), during the last 200 years. I agree that there may be numerous past events of low magnitude ($M_w < 4$), which may not have been detected at all (i.e. especially during the pre-instrumental period and in marine areas), and consequently, they have not been reported in the historical catalogues. Large earthquakes of moderate to high magnitudes ($M_w > 4$) are often identified in the historical catalogues, even if they were located in marine areas. (now in lines 283-288)

L269-L274: I suggest you lower the tone. There are already too many catastrophe sellers.

Thanks a lot for your wise comments. We removed two adjectives that are not essential, “*extensive*” and “*highly*”, and added the word “*eventually*”. (now in lines 285-291)

Reviewer #2 (Remarks to the Author):

Remarks to the author:

This is a very interesting, well written, and well documented research paper. It combines existing knowledge on tectonic evolution with analysis by the authors of various new data sets and presents a convincing case of active left-lateral fault systems, occurring from south to north across the Alboran Sea, that line up to form the Al-Idrissi Fault System (AIFS). With various seismic sections and the aftershocks of the large 2016 earthquake the authors demonstrate that the AIFS cuts through the continental crust of the Alboran basin. The lithosphere of the strongly stretched Alboran basin is rather thin and, being above the subduction wedge for 10s of Myr, it is also expectedly warm. Therefore, the authors could have safely speculated that the AIFS likely starts cutting through the entire Alboran lithosphere. The authors present a convincing case of a nascent continental fault system that defines the plate boundary between Europe (to the east and north) and Nubia (to the west and south). For long, the location of this plate boundary has been enigmatic and subject of debate but with their findings the authors provide the compelling evidence that the AIFS acts as the plate boundary across the Alboran. This finding is a crucial step forward for unraveling this complex tectonically complex region and is provides a clear direction for further investigations of the cause of earthquakes in the region for which the authors present the interesting case that the magnitude of earthquakes along the AIFS may increase to larger than M7 once the AIFS is entirely broken.

The focus of the paper is much on the crustal structure and earthquake generating potential associated with the AIFS. The identification of the regional plate boundary fault system is equally exciting news. The paper substantiates a clear direction for the important issue of where the plate boundary is in the region which not only gives guidance to research into the seismo-tectonic but also to the past tectonic evolution of the region.

This latter aspect is, however, a bit undertreated by the authors and I would like to suggest to briefly review (in the Introduction) the various literature propositions made for the tectonic role of the TASZ. It has only been propounded as plate boundary by some and not by most researchers, e.g. see Vernant 2010 for several literature propositions. The Bird-proposition of a plate boundary running from the Atlantic through northern Morocco seems to be mostly used regionally (e.g. Neres et al. 2016) as well as in the latest global plate kinematic models for the last few million years (Argus et al. 2011) as successor of the NUVEL model. I suggest to enhance this discovered role of the AIFS, which is now only touched upon in the last few sentences of the paper (271-274).

I have only few minor comments and advise publication after minor revision.
Wim Spakman

Dear Wim, thank you for your positive and constructive comments. We really appreciate them, and we are sure they will improve the content of our paper.

E.G: Thanks for your suggestion. In the Introduction, we briefly reviewed the three literature propositions regarding the tectonic role of the TASZ

Thanks for your suggestion. This is how it is now included in the text of the ms:

“The TASZ is traditionally interpreted as a complex belt of deformation that crosscuts the Alboran Sea and its two margins, connecting the Rif (North Africa) to the Eastern Betic Shear Zone (SE Iberian Peninsula)^{18,19}. Only a few works have proposed that the TASZ may play the role of a plate boundary across the Alboran Sea, traversing the Nubia-Eurasia plates in the westernmost Mediterranean^{20,21,22}”.

The suggested articles are now added to the reference list:

- 20. Vernant, P. et al. Geodetic constraints on active tectonics of the Western Mediterranean: Implications for the kinematics and dynamics of the Nubia-Eurasia plate boundary zone. *Journal of Geodynamics* 49, 123-129 (2010).
- 21. Argus, D.F., Gordon, R.G., & DeMets, C. Geologically current motion of 56 plates relative to the no-net-rotation reference frame. *Geochem. Geophys. Geosyst.* 12, Q11001, doi:10.1029/2011GC003751 (2011).
- 22. Neres, M. et al. Lithospheric deformation in the Africa-Iberia plate boundary: Improved neotectonics modelling testing a basal-driven Alboran plate. *J. Geophys. Res.* 121, 6566-6596, 2016.

(now in lines 57-62)

Title: replace ‘embryonic’ by ‘incipient’ (it is beyond embryonic’)

I fully agree and we thank you for the comment. We have replaced “*embryonic*” with “incipient” in the title as well as the Discussion section.

Continental -> ‘plate boundary’; this is more to the point while ‘continental’ is rather general. Although the Alboran comprises strongly stretched continental crust, it can hardly be taken as typical for ‘continental’

37: continental -> plate-boundary

We agree with your proposition, thank you. In the abstract, we changed “*continental*” to “plate boundary”.

42-43: Crustal deformation cannot be ‘mainly’ explained by relative plate convergence (ref 14). Suggest to replace by: ... fault systems within the overall plate tectonic setting of NW-SE trending convergence (Fig. 1).

Thank you. We have modified the text of the manuscript based on your suggestion.

(now in lines 43-44)

51 ref 13 (Wortel and Spakman 2000) is not appropriate because they did not state that. Better replace by Gutscher 2014 and/or Spakman and Wortel 2004.

Thanks for your suggestion. We removed the previous reference number 13 of “*Wortel and Spakman, 2000*”, and changed it to the new reference now number 16 “*Spakman and Wortel, 2004*”, as proposed. **(now in line 56)**

58: “plate subduction activated” is vague. Suggestion: “ ... by NNE directed dragging of the RGB slab by the Nubia plate in the past ...’ This also improves the flow to the next sentence.

We appreciate your suggestion. We changed the previous sentence to: “...induced by NNE directed dragging of the RGB slab by the Nubia plate in the past 8 Myr,”
(now in line 65)

64-68: This interrupts a bit the flow of the introduction. Suggest to move to line 49, or to provide a bridge with the previous paragraph.

We agree and thank you for your wise suggestion. We have moved the paragraph of lines 64-68 to line 49, and changed the references accordingly. **(now in lines 49-54)**

68 ‘map’ -> ‘structural map’

Thanks for the reminder. Actually, we now realize that we are not just referring to a map or a structural map, but to a “*new and comprehensive geological and geophysical dataset*” of the entire AIFS. **(now in line 71)**

77 ‘might’ -> ‘may’

OK, thanks. We have changed “*might*” to “*may*”. **(now in line 79)**

81 “NE-SW” -> “NNE-SSW”; the azimuths you report later belong to NNE-SSW

We completely agree. We have changed the azimuths to NNE –SSW, as you suggested. **(now in line 83)**

247 Boundaries between plates are defined by lithosphere cutting faults, not by crustal boundaries. You could make the case here that the AIFS may as well cut the thin hot Alboran lithosphere (likely by fault creep).

Thank you for your last comment, we agree with you. Accordingly, we have modified the sentence as follows: “*The AIFS... is growing and ... could develop into a large-scale continental plate-boundary fault along the Trans-Alboran Shear Zone*”. **(now in lines 264-266)**

....*all these systems form major lithospheric cutting faults...***(now in line 269)**

Reviewer #3 (Remarks to the Author):

Review Gràcia et al., Nature Communications

As the authors say, the occurrence of this intense earthquake sequence on a young continental fault is a great opportunity to understand the evolution of strike-slip fault systems. Besides the treatment of a hot topic, the study is based on excellent data from different bathymetric and seismic campaigns, as well as earthquake recordings. The conclusions are well-thought and consistent, interesting for a broad readership, and clearly worth a place in a top scientific journal. I recommend publication in Nature Communications after minor revision.

In the following I will list a number of criticisms and comments to the authors. Some of them should be definitely addressed, I believe, while others may be more optional and arise from my subjective point of view. Some aspects of this study appear less solid than others, and I think the authors might try to clarify or omit these points, to avoid that their general conclusions may become vulnerable, too.

1) The article is constructed around the characterization of the Al Idrisi Fault (AIF) as an embryonic, initial-stage, young, or nascent fault system. I think this is a very appealing and useful model that may explain many features of the region, and it sufficiently justifies the special relevance of analysing this fault. However, some numbers, for example the accumulated slip, may be relevant to explain this concept. For example, the Alboran Ridge (as a bathymetric feature) appears offset by about 10 km by the fault (Fig. 2), which would be consistent with the age (Late Pliocene to Quaternary, e.g. 3 Ma) and plausible slip rates (e.g. 3 mm/a). This, by the way, is a rather normal ratio between fault length and offset. So it seems ok to call the fault initial-stage or young, but embryonic or nascent may be exaggerated.

Thank you for your wise and very interesting comments. We agree with you in that the accumulated slip is relevant for analyzing the AIFS, and that the ratios between fault length and slip dates are “normal” for a young fault in an initial-stage.

Coinciding with other reviewers, I completely agree with your suggestion. We have decided to call the AIFS an “incipient” fault system, as suggested by the 2nd reviewer.
now in line 1 and line 264

2) You predict that the AIF will develop into a large-scale continental plate-boundary; I think also this point deserves more explanation: The AIF is trending close to N-S, approximately perpendicular to the overall direction of the Nubia-Iberia plate boundary. So please describe the trace of the plate boundary that would involve the AIF, and the way that tectonic deformation at this boundary would work.

The AIFS is a left-lateral strike-slip fault system that trends NNE-SSW. This trend is just the conjugate of the NNW-SSE Nubia-Iberia plate convergence (4-5 mm/a), as can be observed in the inset of Figure 7. The tectonic deformation that would work at this boundary is as follows:

- The NNE-SSW trending faults are left-lateral strike slip faults (i.e. AIFS)

- The ENE-WSW trending faults are reverse/thrust faults (i.e. Alboran Ridge FS)
- The WNW-ESE trending faults are right-lateral strike slip faults (i.e. Yusuf FS)
- The NNW-SSE trending faults are normal faults (i.e. the NS Faults)

In the inset of Figure 7 we show the orientation of the different faults in response to the plate convergence.

inset of Figure 7

3) Previous work on the AIF in seismic and bathymetric studies is acknowledged properly and well embedded in the manuscript, but previous seismological studies are not described with sufficient detail. Bufoin et al., 2017 and Kariche et al., 2018 are merely mentioned once in the introduction. These studies have significant overlap with your work, and you should compare data, methods and results.

We agree that we did not describe the previous seismological studies in enough detail, such as Bufoin et al., 2017 and Kariche et al., 2018. Clearly, there is some overlap between our studies. In the methods section we now discuss this issue. Please note that the slip inversion of Bufoin et al. follows the same technique for the rupture, and indicates a 20-25 km long slip patch and rupture in the same direction.

now in lines 202-203

For the Coulomb stress modelling, we prefer our own analysis, as our study is based on bathymetry and seismic information on fault orientation, while the Kariche et al. paper is based on a conceptual orientation of faults in the area.

now in lines 203-204

4) Not enough information is provided on regional moment tensor inversion regarding stations (number of stations, distances, azimuthal gap), earth model (no information at all) and results (please show some waveform fits for some examples of the smaller earthquakes as supplementary material).

Thanks for your comment. Now, for each moment tensor there is a map like the one in **Supplementary Figure 1**. These figures show the stations used, with graphic information of the distances and gaps. The earth model is included in the Methods section **(model=WUS, Western US)**; and for each event, there is a figure like 8c with waveform fits. **now in lines 198 (Sup. Fig. 8a) and 387 (model WUS)**

Finally, you can download the data repository at:

<https://www.dropbox.com/s/fs9v3xm578n2h30/RCMT.zip?dl=0>

When you unzip it (compressed it takes up 4 GB), you will see that there are two files: README.txt and HTML file index.html. Please, read the README file and click on index.html. It should open in a browser, and you can click on each event and see the corresponding solution.

5) The result from finite fault modelling is a strongly asymmetric rupture that propagates about 20 km southwards, at low rupture speed of 2 km/s. This is the opposite from the result proposed in Bufoin et al., an asymmetric rupture that propagates about 20 km northwards, at high rupture speed of 3 km/s. This dissonance is noteworthy

because both studies use the same method and inversion code (Kikuchi and Kanamori), same type of data (teleseismic P and SH waves from IRIS) and also similar models (both incorporating the water layer, which is good). Since Buform et al. is the first study, it corresponds to this study to comment on these differences. Unfortunately, Buform et al. do not provide waveform fits. On the other hand, waveform fits in this study (supplementary Figure 2) do not support the proposed rupture model convincingly. In particular, the longest observed SH pulses (MBO) and P pulses (DBIC, I'd say) are located towards south, which seems to be at odds with the proposed southward propagating rupture. The northwards propagation by Buform et al. is supported by the directivity function of long-distance Rayleigh waves. In this study, a weird attempt to characterize directivity from regional data has been done (supplementary figure 4). The proposed azimuth to make the cosine function fit the data is 0 degrees, i.e. north. This actually fits the shorter arrivals towards north, contradicting the southwards rupture propagation just inferred one paragraph before from teleseismic modelling. My personal balance of evidence is that rupture likely propagated to the north. My suggestion would be to omit this point, because it is really weak, and actually not necessary for the main conclusions of this study. If you want to keep rupture directivity, then please also report what exactly is shown in supplementary figure 4 (Vertical P-waves?), and report the parameter you use to fit the equation given in the figure caption.

Please excuse us for causing confusion. There was a typo in the manuscript between “Northwards” and “Southwards”. As you can see in our Supplementary Material, our rupture propagated northwards (in Supplementary Figure 4 the arrows outlining slip orientation point to the north and NOT towards the south), and hence there is no mismatch between stations MBO and DBIC. Please note that Supplementary Figure 5 also indicates a northwards propagation rupture. Therefore, the two studies on rupture directivity support each other. We have now corrected the typo in the manuscript.
now in lines 177-178

6) I'm really unable to identify the Moho in supplementary figure 1 (**now Supplementary Figure 2**), and in consequence the Moho step across the AIF, so I cannot reach to the conclusion that two different crustal domains are observed across the AIF. In particular, energy vanishes around 10.5 s TWT along the entire profile, not only the SE part. I suppose that this is due to the representation, and you can see the Moho in the original, full-resolution seismic section. Is there any other study that supports the Moho step? Receiver functions from nearby coastal stations (Mancilla and Diaz, Tectonophysics, 2015) rather suggest it's the other way round: The crust is much thinner to the east of the intersection between the AIF and the Moroccan margin, than to the west. Please explain.

The interpretation of the crustal domains and the Moho location is not only based on the small image in Supplementary Fig 2, it is based on a much larger dataset of about 3000 km of deep penetration seismic data collected, processed and interpreted by our group. The results are presented in the PhD of Laura Gomez de la Peña. The part concerning the crustal domains and Moho geometries has been recently published in *Tectonics* (2018). In this article we present large scale figures of numerous seismic profiles that support our interpretation:

Gómez de la Peña, L., Ranero, C.R., Gràcia, E. The crustal domains of the Alboran Sea (Western Mediterranean). *Tectonics*, **37**, doi:10.1029/2017TC004946 (2018).

now in lines 461-462

The Mancilla & Diaz paper (2015) uses land-stations and one station located on Alboran Island, at the center of the Alboran Basin. Therefore, for the Moho they do not have constraints offshore. In addition, in the El Moudnib, Villaseñor et al. paper (*Tectonophysics*, 2015), which has local earthquake tomography and much better ray coverage of the offshore region, they found that the crust in the west Alboran Basin is thinner than 5 km or absent.

El Moudnib, L., Villaseñor, A., Harnafi, M., Gallart, J., Pazos, A., Serrano, I., Córdoba, D., Pulgar, J.A., Ibarra, P., Himmi, M., Chour, M. Crustal structure of the Betic–Rif system, western Mediterranean, from local earthquake tomography, *Tectonophysics* **643**, 94-105, 2015 now referred in lines 536-537

Two minor comments:

Line 82: The AIF “accommodates a total rate of ~3.8 mm/yr of relative motion”. This appears speculative, since this number is taken from a GPS study, but the AIF is an offshore fault along its entire length. Please phrase it more carefully.

Considering that the AIFS is the major fault structure in a large region, we infer that it accommodates most of the total rate of 3.8 mm/yr.

now in lines 84-85

Line 332: slip/dip/rake of 124/84/175 and 214/85/5 are not conjugate faults (at ~60 degrees, for typical friction parameters in geology), but the orthogonal fault and auxiliary planes of the symmetric point source mechanism (at 90 degrees).

Sorry, we made a mistake. We have now modified the sentence and changed “conjugate” to “nodal plane”.

now in lines 349-350

Best wishes, E. Gràcia et al.

** See Nature Research’s author and referees’ website at www.nature.com/authors for information about policies, services and author benefits. This email has been sent through the Springer Nature Tracking System NY-610A-NPG&MTS

Confidentiality Statement:

This e-mail is confidential and subject to copyright. Any unauthorised use or disclosure of its contents is prohibited. If you have received this email in error please notify our Manuscript Tracking System Helpdesk team at <http://platformsupport.nature.com>.

Details of the confidentiality and pre-publicity policy may be found here <http://www.nature.com/authors/policies/confidentiality.html>

Privacy Policy | Update Profile

Reviewers' comments:

Reviewer #3 (Remarks to the Author):

I'm happy this paper had been well received by all reviewers, and the authors provided a clearly improved second version of the manuscript. I'm overall satisfied also with the answers to my own questions and comments, and the way they have been integrated in the manuscript (where appropriate), except for two points that still need clarification:

1) In the answer to question 4 (details about MT inversion), you refer to the methods section for information about the Earth model. There you mentioned a Western US model, giving links to Fig. 5, Supplementary Table 1 and reference 64. Neither Fig. 5, nor Table S1 show an Earth model, and reference 64 is about the ak135 model by Kennett et al., please put this right (and show the model in Fig. 5 or Supplementary Table 1).

2) Answer to question 5 (direction of rupture directivity): I'm glad I could help to sort out a problem here; and you actually confirm my guess that rupture propagates northwards, in agreement with pulse widths in teleseismic and regional records. Still, this is apparently inconsistent with Fig. 4. The arrows in Supplementary Fig. 4 show the direction of slip (rake), NOT the (northward) direction of rupture propagation, as you insinuate in the caption. If it was rupture propagation, the arrows would be perpendicular to the instantaneous rupture fronts and we would see a divergent pattern, which is not the case. I guess these arrows show (following the usual conventions for the representation of slip models) the relative motion of the hanging wall (or front block, in case a vertical fault), i.e. the arrows tell us we have fault motion close to left-lateral strike-slip everywhere. They don't tell us about directivity. Directivity in this case can be inferred from the distribution of slip with respect to the hypocentre (star): The moment centroid (~center of mass of slip distribution) is clearly south of the hypocentre. This, however, indicates predominately SOUTHward rupture directivity. What's wrong?

Reviewer #4 (Remarks to the Author):

REVIEW OF THE MS NCOMMS-18-35440A BY Gràcia and co-authors

General comments:

The paper by Gràcia and co-authors combines newly acquired sea-floor data with seismological analysis to evidence, for the first time and in a very robust and convincing way, the presence of a major tectonic structure within the Al-Idrissi Fault System that is commonly considered as a complex diffuse plate boundary.

I find that the paper is well structured, benefits from high quality figures and, most importantly, it improves the scientific knowledge in a complex region that was responsible for large earthquakes in the past. I believe that this paper will hopefully make significant impact after publication and contributes to increase the awareness to the earthquake and associated hazards (tsunami and landslide) in the Alboran Sea region.

As the manuscript already underwent one round of review, it is worth to mention that the authors addressed all the reviewers' comments and accordingly inserted their suggestions and corrections. Overall, a good paper but needs a bit of work to make it an excellent paper. I recommend the publication of the paper, but, in my opinion, it needs one good read-through and revision by a native English speaker/writer to smooth out some of the wording.

Specific comments/suggestions:

Here are some specific examples:

1. The authors used different units for the same quantity: mm.yr⁻¹ (Line 45), mm/yr (Line 87), and mm/a (Line 272). This must be uniform over the manuscript.
2. For the sake of consistency, when quantifying the faults dimension use the same expression "...-km-long/wide". Please have a read through and fix any others that may be in there.
3. Line 49 -50: This is awkward, please rephrase. How about: "This last event caused 626 fatalities and left 5,600 homeless, making it the most catastrophic earthquake in the region during the last century."
4. Lines 73-77: In fact, the authors' multi-scale approach doesn't consist of data sets, as mentioned here, but of the methods used to analyse and interpret these data sets. I suggest: "We adopted a multi-scale approach, including detailed morphological analysis of shipboard multibeam bathymetry and near-bottom bathymetry obtained with Autonomous Underwater Vehicles (AUVs), and interpretation of deep penetration multichannel-seismic (MCS) data (Figs. 2a and 3)."
5. Lines 87-94: Again, just awkward wording and a too long sentence hard to read and understand. Please rephrase and split it in two sentences.
6. Line 100: allow identifying instead of "make it possible to identify".
7. Line 136: we provide instead of "we give"
8. Line 141: delete "well"
9. Line 145: (North AIFS segment) instead of "(North AIFS)"
10. Lines 147-148: allows proposing instead of "makes it possible to propose"
11. Lines 158: add the word segment after AIFS.
12. Lines 168-173: This whole section needs better referencing.
13. Line 175: The coordinates often present the location of the event epicentre. Please try: " For the mainshock, we located the epicentre at 35.59°N and 3.72°W,...."
14. Line 193: add the word segment after AIFS.
15. Lines 205-206: For consistency, replace "Al-Idrissi Fault System" by its acronym AIFS.
16. Line 210: delete magnitude before Mw 6.4.
17. Lines 210-211: please reword. Try "...an Mw 6.4 earthquake would require the rupture of at least a 30-km-long fault."
18. Line 214: add the article "the" before profile TM1.
19. Lines 220-222: I didn't get here the aim of modelling the change in Colomb failure stress. Is it to show that the occurrence of large earthquake in the region could activate the AIFS?. Please clarify.
20. Line 229: delete "it"
21. Lines 230 and 231: add AIFS after Central.
22. Lines 258 and 261: same event, 1804, with different IO (8-9;7-8)?
23. Lines 273-275: Need references.
24. Lines 281: add "segment" after the South AIFS.
25. Line 282: add "segments" after the North and Central AIFS.
26. Lines 290-293: Needs rewording. I suggest: "Given the low awareness and preparedness for seismic and tsunami hazards in the region, major earthquake may eventually cause dramatic damage along the highly populated coastal zones of the Alboran Sea. Therefore, large events should be considered in future seismic and tsunami hazard assessments and mitigation plans."
27. Line 314: Just "The MCS profiles"
28. Line 362: Like the case of the January 2016 Alboran Sea earthquake.
29. Line 363: Can only be determined through seismic data instead of "can only be based on seismic data"
30. Lines 369-370: Please rephrase the sentence: "The model assumes that the rupture consists of a propagating rupture front with slip accruing in the wake of the passage of the rupture front.", which is confusing.
31. Lines 460-461: The authors claimed the availability of data by they didn't mention where
32. Maps are generally very good. Please just add the North arrow in Fig 1, Fig 2, Fig 4a and Fig 6.

Rachid Omira

Reviewers' comments:

Reviewer #3 (Remarks to the Author):

I'm happy this paper had been well received by all reviewers, and the authors provided a clearly improved second version of the manuscript. I'm overall satisfied also with the answers to my own questions and comments, and the way they have been integrated in the manuscript (where appropriate), except for two points that still need clarification:

1) In the answer to question 4 (details about MT inversion), you refer to the methods section for information about the Earth model. There you mentioned a Western US model, giving links to Fig. 5, Supplementary Table 1 and reference 64. Neither Fig. 5, nor Table S1 show an Earth model, and reference 64 is about the ak135 model by Kennett et al., please put this right (and show the model in Fig. 5 or Supplementary Table 1).

Thanks for your comments. Here are our answers:

The WUS model is described in Herrmann et al., 2011 (BSSA), which corresponds to reference number 64. Specifically, the model can be found in Figure 3 of the Herrmann's (2011) paper.

Below, there is a table with specific parameters of the WUS model. The relevant columns are H(KM) VP(KM/S) VS(KM/S) RHO(GM/CC).

Important, H correspond to layer thickness, not depth. So to list depth in the table we need to add the thickness. The last value of H is 0.0, meaning that it is an infinite half space (below Moho the model is homogeneous)

Herrmann, R. B., Benz, H. M., & Ammon, C. J. (2011). Monitoring the Earthquake Source Process in North America. *Bulletin of the Seismological Society of America*, 101(6), 2609–2625. <http://doi.org/10.1785/0120110095>

H(KM)	VP(KM/S)	VS(KM/S)	RHO(GM/CC)	QP	QS	ETAP	ETAS	FREFF	FREFS
1.9000	3.4065	2.0089	2.2150	0.302E-02	0.679E-02	0.00	0.00	1.00	1.00
6.1000	5.5445	3.2953	2.6089	0.349E-02	0.784E-02	0.00	0.00	1.00	1.00
13.0000	6.2708	3.7396	2.7812	0.212E-02	0.476E-02	0.00	0.00	1.00	1.00
19.0000	6.4075	3.7680	2.8223	0.111E-02	0.249E-02	0.00	0.00	1.00	1.00
0.0000	7.9000	4.6200	3.2760	0.164E-10	0.370E-10	0.00	0.00	1.00	1.00

You can download the data repository at:

<https://digital.csic.es/handle/10261/177887>

<http://dx.doi.org/10.20350/digitalCSIC/8623>

2) Answer to question 5 (direction of rupture directivity): I'm glad I could help to sort out a problem here; and you actually confirm my guess that rupture propagates northwards, in agreement with pulse widths in teleseismic and regional records. Still, this is apparently inconsistent with Fig. 4. The arrows in Supplementary Fig. 4 show the direction of slip (rake), NOT the (northward) direction of rupture propagation, as you insinuate in the caption. If it was rupture propagation, the arrows would be perpendicular to the instantaneous rupture fronts and we would see a divergent pattern,

which is not the case. I guess these arrows show (following the usual conventions for the representation of slip models) the relative motion of the hanging wall (or front block, in case a vertical fault), i.e. the arrows tell us we have fault motion close to left-lateral strike-slip everywhere. They don't tell us about directivity. Directivity in this case can be inferred from the distribution of slip with respect to the hypocentre (star): The moment centroid (~center of mass of slip distribution) is clearly south of the hypocentre. This, however, indicates predominately SOUTHward rupture directivity. What's wrong?

Thanks for your explanation and comment. You are right, the slip is not directly telling us anything about directivity. We should correct the caption.

There is, however, nothing "wrong" with neither the slip nor the directivity. Indeed, in the figure showing the slip there is an offset or bias between centroid and hypocenter. It is caused by some bias between the estimate of the hypocenter derived from "local earthquake analysis" and "teleseismic analysis for the slip inversion". The local assessment for the hypocentre is based on our velocity model for the Alboran Sea and it's correction for 3D velocity effects occurring within the Alboran Sea, Rif and Betics (using regional arrival times from the IGN catalogue). The teleseismic slip inversion (and inversion for the centroid) is using a global 1-D velocity model (AK135) and seismic stations at much larger teleseismic distances. Differences in both procedures have indeed caused some offset between centroid and hypocenter. Thus, for the teleseismic inversion, any heterogeneity between the source and receiver may cause small differences in arrival times, which would cause a bias of the centroid location. Without knowing the real 3-D structure of Earth, some bias is inherently related to teleseismic techniques. Consequently, it would have been a miracle if both procedures would provide the same location (note - even the IGN location of the main shock is much more biased than the bias between our hypocenter and the centroid!). We indicate this fact in the revised figure caption.

See below the text of the caption corrected:

Supplementary Figure 4:

*“Coseismic slip (in centimeters) determined from the inversion of the teleseismic waveforms, yielding shallow slip and hence potential surface rupture in the vicinity of the epicenter of the 25 January 2016 earthquake. The **yellow star** indicates the centroid of the point source approximation of the mainshock. Slip direction is indicated by arrows, favoring a northward propagation of the rupture front. Note that the lengths of arrows correspond to slip magnitude.*

In the caption of Supplementary Figure 4, we added the following comment:

“Please note that hypocentre location of the mainshock (yellow star) and centroid depth or maximum of slip are shifted with respect to each other by ~8 km. The offset is controlled by differences in location procedures. Thus, all earthquake hypocentres are derived from regional observations at the IGN network, using a local velocity model and station corrections for the Alboran domain (see Methods). In contrast, slip inversion uses observations at large epicentral distances (30-90°) and a global 1-D velocity model (see Methods)”.

Reviewer #4 (Remarks to the Author):

General comments:

The paper by Gràcia and co-authors combines newly acquired sea-floor data with seismological analysis to evidence, for the first time and in a very robust and convincing way, the presence of a major tectonic structure within the Al-Idrissi Fault System that is commonly considered as a complex diffuse plate boundary.

I find that the paper is well structured, benefits from high quality figures and, most importantly, it improves the scientific knowledge in a complex region that was responsible for large earthquakes in the past. I believe that this paper will hopefully make significant impact after publication and contributes to increase the awareness to the earthquake and associated hazards (tsunami and landslide) in the Alboran Sea region.

As the manuscript already underwent one round of review, it is worth to mention that the authors addressed all the reviewers' comments and accordingly inserted their suggestions and corrections. Overall, a good paper but needs a bit of work to make it an excellent paper. I recommend the publication of the paper, but, in my opinion, it needs one good read-through and revision by a native English speaker/writer to smooth out some of the wording.

Specific comments/suggestions:

Here are some specific examples:

1. The authors used different units for the same quantity: mm.yr⁻¹ (Line 45), mm/yr (Line 87), and mm/a (Line 272). This must be uniform over the manuscript.

Thank you for your suggestion. Now, I have homogenized the units, referring to **mm/yr**.

2. For the sake of consistency, when quantifying the faults dimension use the same expression "...-km-long/wide". Please have a read through and fix any others that may be in there.

We completely agree with your comment. We have now modified the text of the ms. referring to **x-km-long** and/or **x-km-wide**.

3. Line 49 -50: This is awkward, please rephrase. How about: "This last event caused 626 fatalities and left 5,600 homeless, making it the most catastrophic earthquake in the region during the last century."

Thanks for your comment. We have now rephrased the sentence as you suggested, which improve to understand the sentence.

4. Lines 73-77: In fact, the authors multi-scale approach doesn't consists of data sets, as mentioned here, but of the methods used to analyse and interpret these data sets. I

suggest: “We adopted a multi-scale approach, including detailed morphological analysis of shipboard multibeam bathymetry and near-bottom bathymetry obtained with Autonomous Underwater Vehicles (AUVs), and interpretation of deep penetration multichannel-seismic (MCS) data (Figs. 2a and 3).”

I completely agree with your suggestion, thank you! Now the sentence is clearer and the content of the sentence focuses not only on the datasets, but also on the different methodologies that we have used.

5. Lines 87-94: Again, just awkward wording and a too long sentence hard to read and understand. Please rephrase and split it in two sentences.

Thanks for your comment. Here are the new paragraph:

“The AIFS runs from the Djibouti Plateau in the north, where the historical 1804 earthquake occurred (DJ04, MSK Intensity VIII)⁸(Fig. 2a), to the Nekor Basin (Moroccan margin) in the south (Fig. 2b). Towards the North, the AIFS connects to a parallel structure, a wide shear-zone defined as the NS Faults system (NSF)⁵, located near where the destructive 1910 Adra earthquake (AD10, estimated Mw ~6.1)²⁵ and the 1993-1994 seismic crises (AD93-94) occurred^{7,15} (Fig. 2a). To the south, the AIFS links to the Trougout and Bokkoya faults (Nekor Basin), the last of which is related to the source of the Al-Hoceima 1994 earthquake⁸⁻¹⁰ (Fig. 2a, b)”.

6. Line 100: allow identifying instead of “make it possible to identify”.

Thanks. We have changed “make it possible to identify” for **allow identifying**.

7. Line 136: we provide instead of “we give”

Thank you! We have changed we give by “**we provide**”.

8. Line 141: delete “well”

Instead of deleting the term “well”, I suggest to changed it by “**wells**”. A commercial well is any drilling site that produces enough oil or gas to be commercially viable. In the north Alboran Sea, there are several commercial well-sites that helped us to calibrate the seismo-stratigraphy in our study area. In addition, there are also scientific wells from ODP (Ocean Drilling Project) Leg 161 carried out in the area (Sites 976, 977, 978, and 979).

9. Line 145: (North AIFS segment) instead of “(North AIFS)”

Thank you! I have added “**segment**” to the sentence.

10. Lines 147-148: allows proposing instead of “makes it possible to propose”

I have changed the sentence as suggested: “**this allows proposing**”.

11. Lines 158: add the word segment after AIFS.

Thanks. I’ve added “**segment**” after AIFS.

12. Lines 168-173: This whole section needs better referencing.

Thank you, I completely agree. We now refer to the articles related to the 2016 earthquake of magnitude M_w 6.4 including the foreshock and aftershocks until the 13th of May 2016. We also quote the 3 following references:

13. **Buforn, E. et al.** The 2016 south Alboran earthquake ($M_w = 6.4$): A reactivation of the Ibero-Maghrebian region? *Tectonophysics* **712-713**, 704-715 (2017).
14. **Kariche, J., Meghraoui, M., Timoulali, Y., Cetin, E. & Toussaint, R.** The Al Hoceima earthquake sequence of 1994, 2004 and 2016: Stress transfer and poroelasticity in the Rif and Alboran Sea region. *Geophys. J. Int.* **212**, 42-53 (2018).
15. **Instituto Geográfico Nacional (IGN)**, Ministerio de Fomento, Gobierno de España, Madrid. www.ign.es/resources/sismologia (2018).

13. Line 175: The coordinates often present the location of the event epicentre. Please try: “For the mainshock, we located the epicentre at 35.59°N and 3.72°W,....”
We have modified the text as you suggested. Thanks!

14. Line 193: add the word segment after AIFS.
Thanks for the comment, **segment** is added after AIFS.

15. Lines 205-206: For consistency, replace “Al-Idrissi Fault System” by its acronym AIFS.
Thanks again! It is done as you suggested.

16. Line 210: delete magnitude before M_w 6.4.
The term “**magnitude**” has been deleted.

17. Lines 210-211: please reword. Try “...a M_w 6.4 earthquake would require the rupture of at least a 30-km-long fault.”
Thank you for your comment. The sentence has been wrote as you suggested.

18. Line 214: add the article “the” before profile TM1
The article “**the**” has been added.

19. Lines 220-222: I didn't get here the aim of modelling the change in Coulomb failure stress. Is it to show that the occurrence of large earthquake in the region could activate the AIFS? Please clarify.
The aim of the modelling is to illustrate how moderate to large earthquakes a) may control the distribution of aftershocks, and b) could possibly trigger large earthquakes along the AIFS and nearby faults; as mentioned in the ms.

20. Line 229: delete “it”
“**it**” has been deleted

21. Lines 230 and 231: add AIFS after Central.
AIFS has been added after Central. Thanks.

22. Lines 258 and 261: same event, 1804, with different I_0 (8-9;7-8)?

Thanks for your comment, this is a very good question. The fact is that there are 2 historical events during the same year 1804.

A) The first event is located in the **center of the Alboran Sea** at **36°5'N** on the Djibouti Plateau, in a parallel trace of the AIFS North segment. This event occurred between the 13th and 21st January 1804. The maximum intensity was on **13th January 1804 (I_0 7-8)**. See Figure 7 (yellow star).

B) The second 1804 event is located onshore, in the **Campo de Dalías area at 36°45'N on the Iberian Peninsula**. This event occurred between the 23rd August 1804 and 30th September 1804. The maximum intensity was on the **25th August 1804 (I_0 8-9)**. See Figure 7 (yellow star).

Year	Mo	Da	Ho	Mi	Se	Epicentral area	MDPs	Imax
1804	01	13	17	53		Mar de Alborán	21	7-8
1804	01	13	21	03		Mar de Alborán	4	6
1804	01	21	04	53		Mar de Alborán	7	7
1804	08	23	15	45		Dalías	3	6
1804	08	25	08	25		Dalías	30	8-9
1804	08	25	16	45		Dalías	1	6
1804	08	25	18	30		Dalías	1	7
1804	08	28	16	50		Dalías	1	6
1804	08	29	07	30		Dalías	1	6
1804	08	29	12	05		Dalías	2	7
1804	09	01	00	30		Dalías	1	5-6
1804	09	03	09	45		Dalías	1	5-6
1804	09	09	05	30		Dalías	2	6
1804	09	23	03	45		Dalías	1	5-6
1804	09	30	04	30		Dalías	1	5-6

We have modified the sentence as:

“...a series of earthquakes occurred during the last 200 years, including in **two historical events in year 1804 (I_0 8-9 and I_0 7-8) and the instrumental 1910 and 1994 events (Fig. 7c)**”.

23. Lines 273-275: Need references.

Thanks for your comment. The references are now included. They correspond to references number 45 and 57:

45. Wesnousky, S. G. Predicting the endpoints of earthquake ruptures. *Nature* **444**, 358-360, (2006).

57. Wesnousky, S.G. Displacement and Geometrical Characteristics of Earthquake Surface Ruptures: Issues and Implications for Seismic-Hazard Analysis and the Process of Earthquake Rupture. *Bull. Seismol. Soc. Am.* **98**(4), 1609–1632 (2008).

24. Lines 281: add “segment” after the South AIFS.

Thanks for your suggestion. Now the term “segment” has been added to the sentence, which now is: “**South AIFS segment**”.

25. Line 282: add “segments” after the North and Central AIFS.

Thanks. We have added “segments” to the sentence, which now is: “**North and Central AIFS segments**”.

26. Lines 290-293: Needs rewording. I suggest: “Given the low awareness and preparedness for seismic and tsunami hazards in the region, major earthquake may eventually cause dramatic damage along the highly populated coastal zones of the Alboran Sea. Therefore, large events should be considered in future seismic and tsunami hazard assessments and mitigation plans.”

Thanks for improving the sentences (changes are in **bold**). Now the final outcome is: “**Given** the low awareness **and preparedness for** seismic and tsunami hazards^{58,59} in **the** region, major earthquake may eventually cause **severe** damage along **the highly populated coastal zones of the Alboran Sea**. **Therefore, large events** should be considered in future seismic **and tsunami** hazard assessments and **mitigation plans**.”

27. Line 314: Just “The MCS profiles”
Thank you. We have changed the sentence as suggested.

28. Line 362: Like the case of the January 2016 Alboran Sea earthquake.
Thank you. We have added “**the case of the**” in the suggested sentence.

29. Line 363: Can only be determined through seismic data instead of “can only be based on seismic data”
Well done, thanks! We have changed the sentence as you proposed. Now the final sentence is: “**...can only be determined through seismic data**”.

30. Lines 369-370: Please rephrase the sentence: “The model assumes that the rupture consists of a propagating rupture front with slip accruing in the wake of the passage of the rupture front.”, which is confusing.
I agree that the sentence is not very clear. We propose the following one: “The model assumes that the rupture consists of a propagating rupture front with slip **accumulated** in the wake of **the rupture front passage**.”

31. Lines 460-461: The authors claimed the availability of data but they didn’t mention where
Data Availability are in the Repository data_NCOMM, which contains the following datasets:

TOPOGRAPHY:

NASA JPL (2013). *NASA Shuttle Radar Topography Mission Global 1 arc second* [Data set].
NASA EOSDIS Land Processes DAAC.
[doi: 10.5067/MEaSURES/SRTM/SRTMGL1.003](https://doi.org/10.5067/MEaSURES/SRTM/SRTMGL1.003)

BATHYMETRY:

EMODnet Bathymetry Consortium (2018): *EMODnet Digital Bathymetry (DTM)*.
<http://doi.org/10.12770/18ff0d48-b203-4a65-94a9-5fd8b0ec35f6>

We include two files:

F4_2018.xyz
F4_2018_rgb.tif
(see EMODNet-bathymetry.zip file)

MULTICHANNEL-SEISMIC REFLECTION PROFILES:

We include three files:

TM02-cortical.pdf
TM22-cortical.pdf
TM28-cortical.pdf
(see SEISMICS.zip file)

REGIONAL CENTROID MOMENT TENSORS

The repository of regional centroid moment tensors can be downloaded at this link:

<https://digital.csic.es/handle/10261/177887>
<http://dx.doi.org/10.20350/digitalCSIC/8623>

DATA FROM IRIS DATA CENTRE

This work included data from the II, IU, GE, GT, and G seismic networks obtained from the IRIS data centre

<http://dx.doi.org/doi:10.7914/SN/II>; <http://dx.doi.org/doi:10.7914/SN/IU>;
<http://dx.doi.org/doi:10.14470/TR560404>; <http://dx.doi.org/doi:10.7914/SN/GT>;
<http://dx.doi.org/doi:10.18715/GEOSCOPE.G>

DATA FROM THE IGN CATALOGUE

<http://www.ign.es/web/en/ign/portal/sis-catalogo-terremotos> (last visited January 2018).

32. Maps are generally very good. Please just add the North arrow in Fig 1, Fig 2, Fig 5a, Fig 6 and Fig. 7.

Thanks a lot for your suggestion. However, in this case, I do not think that is necessary to add a North arrow to each of the maps referred above (Figs. 1, 2, . All the maps are represented with geographic coordinates (North and West), and thus, is very clear where the North it is located. I've searched other NCOMM articles, and I've seen that several maps are located only with coordinates, so the North arrow is not necessarily required. Thus, I would prefer to leave the figures as they are.

Rachid Omira

propagación del frente de ruptura con deslizamiento acumulado tras el paso del frente de ruptura.

REVIEWERS' COMMENTS:

Reviewer #3 (Remarks to the Author):

The authors provide useful answers to the two points that I felt still needed clarification and include the corresponding modifications in the new version of this manuscript. I have no further comments or criticisms about this version, and recommend publication in nature communications. .